# Exposure of mice to environmentally relevant per- and polyfluoroalkyl substances (PFAS) alters the sperm epigenome

Leah Gillespie[1,2,13], Jacinta H. Martin [1,2,13] ✉, Amanda L. Anderson[1,2], Ilana R. Bernstein[1,2], Simone J. Stanger[1,2], Natalie A. Trigg[1,2,3], John E. Schjenken [1,2], Anne-Louise Gannon[1,2], Shanu Parameswaran[1,2], Shannon P. Smyth[1,2,4], Colin C. Conine [3,5], Reena Desai[6], David J. Handelsman [6], Geoffry N. De Iuliis [1,2], Andrew L. Eamens [7], Matthew D. Dun [8,9], Brett D. Turner[10,11], Shaun D. Roman [12], Mark P. Green[4] & Brett Nixon [1,2]

Per- and polyfluoroalkyl substances (PFAS) are a large group of persistent synthetic chemicals and ubiquitous environmental contaminants. Mounting evidence demonstrates that PFAS can bioaccumulate and induce adverse health outcomes, including compromising male reproduction. Despite this, the mechanisms by which PFAS elicits these effects remain unclear. Here, we investigate how an environmentally relevant PFAS cocktail impacts the reproductive function of male Swiss CD1 mice. Following twelve weeks of continuous exposure, we collected blood samples for hormone and PFAS quantification and processed reproductive tissues and spermatozoa for histological and functional assessment. PFAS exposure significantly reduced the rate of daily sperm production, likely due to decreased circulating testosterone and dihydrotestosterone. Further, PFAS-exposed spermatozoa displayed marked alterations to their small non-coding RNA profile, which were linked to dysregulation of early-embryonic gene expression. Notably, these changes occured without significant alteration in sperm viability, motility, or the ability to undergo capacitation or support embryonic development. These findings provide new mechanistic insight into how PFAS exposure impacts male reproductive health.

Per- and polyfluoroalkyl substances (PFAS) are a family of synthetic chemicals, which possess a hydrophobic tail with either a fully or partially fluorinated carbon chain and a hydrophilic head group[1]. Such detergent-like physical and chemical properties render PFAS heat-resistant, oil and water repellant, and refractory to conversion through either biotic action or abiotic reactions with water, sunlight, and/or air[2,3]. It is these properties that led to the inclusion of PFAS as a key component of a wide variety of consumer and industrial products, including textiles, paper products, plastic food packaging, non-stick cookware, and fire-fighting foams[4–6]. The extensive use of PFAS combined with their inherent stability has resulted in pervasive exposure of humans, animals, and the environment[5,7]. As a specific example, until recently, PFAS were used as the major ingredient in aqueous film-forming foams (AFFF) owing to their proficiency in suppressing flammable liquid fires[1,8]. It follows that large amounts of AFFF were deployed during firefighting and training activities at both civilian airports and military bases around the globe[8,9]. Such facilities are now recognized as significant sites of PFAS contamination[8], with PFAS readily detected in nearby surface-, bore-, and ground-water[10,11]. Since many of these water sources contribute to drinking water[10] or are used to irrigate crops[12], they also represent a significant source of exposure for humans and animals. Further compounding this situation is the propensity of PFAS dispersal through groundwater transport, atmospheric dispersion and deposition, and soil and landfill leaching, which has meant that measurable PFAS accumulation continues to be detected at substantial distances from their initial point of release[5,6]. Unsurprisingly, ingestion of contaminated drinking water and/or the consumption of crops and animals raised on contaminated water rank among the most common routes of exposure among the general population[9,12] alongside those who are occupationally exposed[13,14]. Indeed, individuals

---

living and working in the vicinity of such sites have serum PFAS concentrations greater than ten-fold higher than those of the general population (i.e., 1500 ng/mL vs 100 ng/mL respectively)[13,14].

Although growing attention has been focused on the potential health effects associated with PFAS bioaccumulation, inconsistencies in findings have meant that the evidence remains inconclusive[15,16]. Such inconsistencies are, at least in part, reflective of the challenges of standardizing the numerous variables within an exposure regimen, with factors such as the PFAS profile administered, route, duration and timing of exposure all capable of influencing the assessable outcomes[16]. Moreover, few studies have investigated PFAS exposures that emulate environmentally relevant mixtures and concentrations; instead, most studies have focused on either a single PFAS chemical or a mixture of PFAS chemicals administered at supra-environmental concentrations[15]. Given the omnipresence of PFAS, it has also proven difficult to disentangle whether any health effects are caused by a previous exposure, current exposure, or a result of chronic PFAS bioaccumulation[16,17]. While such limitations are by no means unique to the study of PFAS toxicology, these challenges make it extremely difficult to assign causality between PFAS exposure and health outcomes in both epidemiological and preclinical studies. This situation is exemplified by epidemiological evidence of potential links between PFAS exposure and dysregulation of male reproductive health, including elevated instances of testicular cancer[13,18] and reduced semen quality[19–24]. Such data identify the male reproductive tract as a site of vulnerability to PFAS exposure, possibly reflecting its abundant expression of fatty acid-binding proteins[25] that have a propensity to bind and sequester PFAS[26]. This is particularly concerning not only in the context of compromising the fertility status of the exposed individual, but also because of the attendant elevated potential for germ-line transmission of disease risk to the next generation. Moreover, male fertility and more specifically, the clinical assessment of basic sperm parameters, represent readily accessible biomarkers of PFAS exposure and a potentially powerful untapped resource with which to predict the long-term health of an individual[27].

To begin to explore this potential, this study was designed to investigate the direct pathophysiological impact of an environmentally relevant cocktail of PFAS[28] on the reproductive health of male mice. For this purpose, adult male mice were administered PFAS via their drinking water for twelve consecutive weeks. At the conclusion of this period, the mice were euthanized, and blood was collected for hormone and PFAS analysis. Reproductive tissues and spermatozoa were also harvested for histological and functional assays of sperm development and function. The major findings of this study demonstrate that, despite significant bioaccumulation in the blood plasma and testes of exposed animals, consumption of the PFAS cocktail did not elicit overt cytotoxicity or genotoxicity within either the somatic or germ cells of the male reproductive tract. PFAS exposure did, however, reduce the circulating concentration of androgenic sex steroids (testosterone and dihydrotestosterone (DHT)) and testicular cholesterol levels, which appeared to correlate with reductions in the rate of daily sperm production (DSP) and the weight of seminal vesicle fluid relative to that of overall body weight. Further, the mature spermatozoa of exposed males presented with an altered small noncoding RNA (sncRNA) profile that appeared to instigate aberrant gene expression at the 4-cell embryo stage. Collectively, these data indicate that paternal PFAS exposure may have significant consequences beyond that of the exposed father to include the development and health of their offspring.

## Results

### PFAS accumulates in the blood plasma and testes of exposed mice

Following the twelve-week PFAS exposure regimen (Fig. 1), samples were analyzed to determine the concentration of PFAS circulating in the plasma and bioaccumulation in the testes (Table 1). None of the nine PFAS administered in this study recorded detectable levels in the control cohort. By contrast, five of the nine administered PFAS (i.e., perfluorooctane sulfonic acid (PFOS), perfluorohexane sulphonic acid

(PFHxS), perfluoroheptane sulfonic acid (PFHpS), perfluorooctanoic acid (PFOA), and perfluoropentane sulfonic acid (PFPeS)) were readily detected in the plasma of male mice exposed to the low PFAS regimen at concentrations far exceeding the PFAS concentration of the administered drinking water (Table 1). With the exception of PFPeS, an equivalent profile of PFAS was also detected in the testes of animals administered the low PFAS cocktail. While the concentrations of PFAS detected in the testes were below that of the plasma, they were nevertheless usually greater than that of the PFAS concentration of the administered drinking water.

Equivalent trends of PFAS bioaccumulation were noted in the plasma and testes resulting from administration of the high PFAS cocktail (Table 1). The higher dose led to consistent detection of all administered PFAS; with the exception of perfluoropentanoic acid (PFPeA) in the plasma and perfluorohexanoic acid (PFHxA) and PFPeA in the testes. More specifically, the concentrations of PFAS recorded in the plasma and testes of exposed animals scaled proportionally with the amount of exposure, such that mice administered the high dose PFAS cocktail (containing ~10× higher concentration of each PFAS) presented with ~11- to 17-fold more of each detectable PFAS compared with those mice administered low dose PFAS (Table 1). These data encouraged an assessment of the propensity of each PFAS to bioaccumulate in the blood and testicular tissue of exposed animals. For this purpose, we calculated the bioaccumulation factor (BAF; a ratio of the concentration of the analyte present in the animal vs that of dietary consumption) for each PFAS. Broadly, this strategy revealed similar BAF values for both the low and high PFAS, inferring that neither dose had approached saturation thresholds in either of the biota matrices examined. By way of example, PFOS was calculated to have BAF values of 11.29 and 10.31 in the blood plasma of mice administered the low and high-dose PFAS cocktails, respectively. In the testes, an equivalent trend was identified whereby PFOS had calculated BAF values of 3.65 and 3.62 for the low and high-PFAS-exposed mice, respectively. Equivalent trends were upheld across all administered PFAS for which BAF values could be calculated, leading us to conclude that, while proportionally more PFAS accumulated in the blood of exposed animals, the testes were indeed prone to PFAS sequestration. Interestingly, in comparing the levels of PFAS administered in this study, formulated to mimic environmentally relevant concentrations[29], to the safe drinking water guidelines reported in Australia (NHMRC), the US Environmental Protection Agency (EPA) and the European Union (EU) (Table 1), it is clear that the administered PFAS cocktails were above these recommended threshold values[30–32].

### PFAS exposure does not affect whole body or tissue weights

Notwithstanding the substantial bioaccumulation of the administered PFAS (Table 1), an analysis of whole-body weight, as well as absolute and relative weights of male reproductive tissues, revealed minimal impact (Fig. 2). Specifically, neither low nor high dose PFAS exposure significantly altered body weight (Fig. 2A) or the combined weights of the testes, epididymides, seminal vesicles or seminal vesicle fluid of exposed males (Fig. 2B–E) in comparison to controls. Accordingly, the weight of the testes, epididymis and seminal vesicles relative to that of whole-body weight were also similar to controls (Fig. 2F, G). However, mice in the high PFAS exposure group did display a significant reduction in seminal vesicle:body weight ratio in comparison to the control ($p = 0.04$; Fig. 2H) and low dose PFAS ($p = 0.03$; Fig. 2I) groups. A similar trend was also observed with the seminal vesicle fluid:body weight ratios whereby the high dose PFAS exposure group was significantly reduced compared to that of control and low dose PFAS exposure groups ($p = 0.0001$ and $p = 0.02$, respectively; Fig. 2I).

### Daily sperm production is reduced by PFAS exposure

The bioaccumulation of PFAS in testis samples (Table 1) prompted our assessment of the impact of PFAS exposure on testis function. This analysis was initiated with a focus on germ cell development, a strategy that revealed a significant reduction in the rate of DSP[33,34]. Notably, however, this response was limited to those males exposed to the low dose PFAS (control vs low; $p = 0.02$) (Fig. 3A) and was not recapitulated among the cohort of

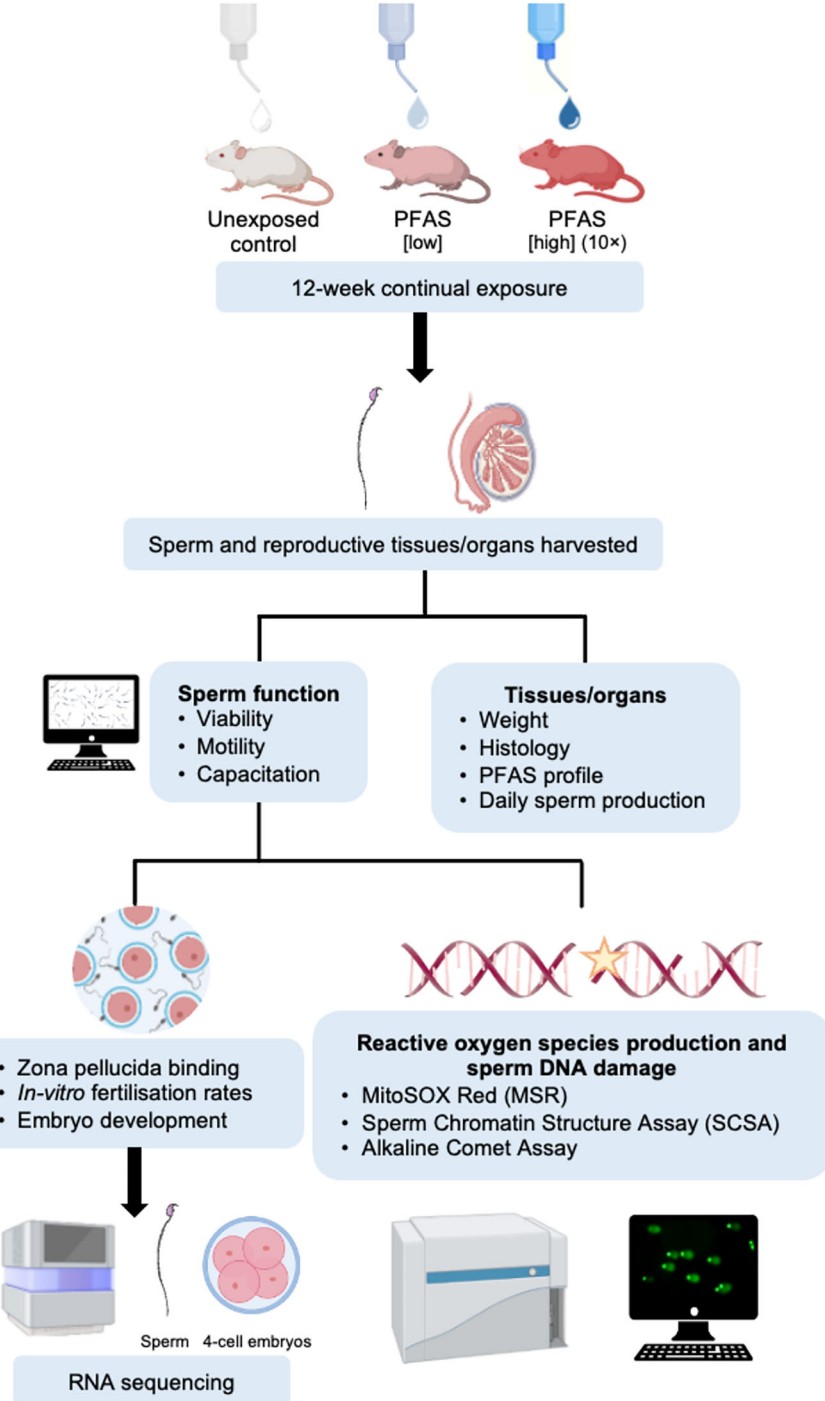

**Fig. 1 | Experimental design.** A total of 216, 4 to 5-week-old Swiss CD1 male mice were exposed to a mixture of nine per- and polyfluoroalkyl substances (PFAS) via their drinking water at environmentally (low) or occupationally (high) relevant concentrations for 12 consecutive weeks. Following exposure, mice were culled, weighed, and prepared for the collection of blood plasma and reproductive tissues/ organs (epididymis, testes, and seminal vesicles). Testes were processed for histological examination and determination of the rate of daily sperm production (DSP). Alternatively, mature sperm cells were isolated from the cauda epididymides and subjected to a suite of functional assays, including viability, motility (via computer assisted sperm analysis (CASA)), ability to capacitate, bind zona pellucidae, fertilize oocytes, and support pre-implantation embryonic development in vitro. Complementing these studies, sperm integrity was assessed by monitoring of the burden of mitochondrial ROS generation (MitoSOX red (MSR)), DNA fragmentation (sperm chromatin structure assay (SCSA)), DNA strand breaks (alkaline comet assay), and dysregulation of their small non-coding RNA profile. Transcriptomic analysis of 4-cell embryos fertilized by PFAS-exposed sperm was also undertaken. Created in BioRender. Nixon, B. (2025) https://BioRender.com/9gzdtdd (Agreement number: EP28OPE772).

male mice exposed to high dose PFAS (Fig. 3A). Gross morphological assessment of hematoxylin and eosin-stained testis sections failed to link this response to overt histopathological changes in either the seminiferous tubules or surrounding interstitial tissue (Fig. 3B; see inserts and white arrowheads). Similarly, analysis of testicular cell apoptosis also failed to identify increases in the proportion of terminal deoxynucleotidyl trasferase (TdT)- mediated dUTP nick end labeling (TUNEL) positive germ or somatic cell populations that could account for deficits in DSP among PFAS exposed males (i.e., TUNEL positive control vs low PFAS $p = 0.62$; control vs high PFAS $p > 0.99$) (Fig. 3C, D).

**Table 1 | Measurement of per- and polyfluoroalkyl substances (PFAS) administered in drinking water and detected in the blood plasma and testicular tissue of exposed mice**

| Chemical name | Abbreviation | Chemical formula | Safe drinking water guidelines[a,b] NHMRC (μg/L) | EPA (μg/L) | EU (μg/L) | Control[c] Water sample (μg/L) | Blood plasma (μg/L) | Testis (μg/kg) | Low PFAS-exposed[c] Water sample (μg/L) | Mouse plasma (μg/L) | Testis (μg/kg) | High PFAS-exposed[c] Water sample (μg/L) | Mouse plasma (μg/L) | Testis (μg/kg) | Estimated mouse consumption[d] Control (mg/kg/day) | Low (mg/kg/day) | High (mg/kg/day) | Estimated human consumption[e] Control (mg/kg/day) | Low (mg/kg/day) | High (mg/kg/day) | Bioaccumulation factor (BAF)[b,f] Low plasma (unitless) | Low testis (unitless) | High plasma (unitless) | High testis (unitless) |
|---|---|---|---|---|---|---|---|---|---|---|---|---|---|---|---|---|---|---|---|---|---|---|---|---|
| Perfluorooctane sulfonic acid | PFOS | $C_8F_{17}\cdot SO_3H$ | 0.008 | 0.004 | 0.1 | ND | ND | ND | 110 | 1242.00±200.41 | 402.00±53.24 | 1300 | 13400.00±1166.19 | 4700.00±989.95 | 1.71E-05 | 0.0135 | 0.1592 | 5.95E-06 | 0.0047 | 0.0553 | 11.29 | 3.65 | 10.31 | 3.62 |
| Perfluorohexane sulfonic acid | PFHxS | $C_6HF_{13}O_3S$ | 0.03 | 0.01 | 0.1 | ND | ND | ND | 32 | 902.00±127.73 | 202.00±41.64 | 310 | 10120.00±1105.07 | 2275.00±517.00 | 7.35E-06 | 0.0039 | 0.0380 | 2.55E-06 | 0.0014 | 0.0132 | 28.19 | 6.31 | 32.65 | 7.34 |
| Perfluorohexanoic acid | PFHxA | $C_6HF_{11}O_2$ | - | - | 0.1 | 0.02 | ND | ND | 15 | ND | ND | 150 | 5.00±0.0* | ND | 0 | 0.0018 | 0.0184 | 0 | 0.0006 | 0.0064 | - | - | - | - |
| Perfluoroheptane sulfonic acid | PFHpS | $C_7F_{15}SO_3H$ | - | - | 0.1 | ND | ND | ND | 8.5 | 174.00±30.43 | 31.00±6.17 | 95 | 2000.00±252.98 | 427.50±99.86 | 1.22E-06 | 0.0010 | 0.0116 | 4.25E-07 | 0.0004 | 0.0040 | 20.47 | 3.65 | 21.05 | 4.50 |
| Perfluorooctanoic acid | PFOA | $C_7HF_{15}O_3S$ | 0.2 | 0.004 | 0.1 | ND | ND | ND | 10 | 80.60±15.28 | 13.78±2.28 | 96 | 882.00±114.12 | 172.50±36.14[g] | 0 | 0.0012 | 0.0118 | 0 | 0.0004 | 0.0041 | 8.06 | 1.38 | 9.19 | 1.80 |
| Perfluorobutane sulfonic acid | PFBS | $C_4HF_9O_3S$ | 1.0 | - | 0.1 | ND | ND | ND | 5.9 | ND | ND | 62 | 11.00±1.95 | 3.00±0.00 | 1.22E-06 | 0.0007 | 0.0076 | 4.25E-07 | 0.0003 | 0.0026 | - | - | 0.18 | 0.05 |
| Perfluoropentane sulfonic acid | PFPeS | $C_5HF_{11}O_3S$ | - | - | 0.1 | ND | ND | ND | 4.6 | 3.80±0.73 | ND | 48 | 65.60±8.96 | 9.50±1.85 | 0 | 0.0006 | 0.0059 | 0 | 0.0002 | 0.0020 | 3.80 | 0.83 | 1.37 | 0.20 |
| Perfluoropentanoic acid | PFPeA | $C_7HF_{19}O_2$ | - | - | - | ND | ND | ND | 3.5 | ND | ND | 38 | ND | ND | 0 | 0.0004 | 0.0047 | 0 | 0.0001 | 0.0016 | - | - | - | - |
| Perfluoroheptanoic acid | PFHpA | $C_6HF_9O_2$ | - | - | 0.1 | ND | ND | ND | 2.7 | ND | ND | 29 | 8.20±1.24 | 2.50±0.50[g] | 0 | 0.0003 | 0.0036 | 0 | 0.0001 | 0.0012 | - | - | - | - |
| Sum total PFAS | | | | | | 0.02 | 0.00 | 0.00 | 192.20 | 2398.60 | 652.60 | 2128.00 | 26486.80 | 7590.00 | 2.69E-05 | 0.02 | 0.26 | 9.36E-06 | 0.01 | 0.09 | 12.48 | 3.40 | 12.45 | 3.57 |

[a]National Health and Medical Research Council (NHMRC)[30], EPA = Environmental Protection Agency Maximum Contaminant Levels[31], EU = European Union Drinking Water Directive[32].

[b] '-' = unable to determine.

[c]ND ≤ Practical Quantitation Limit (PQL). Water PQL = 0.01 μg/L. Plasma and testes PQL = 2 μg/L for all except for PFPeA which is 4 μg/L.

[d]Estimated mouse consumption based on reports by Hedrich et al.[67], in which it is stated that mice consume an average of 6 mL/animal/day and an average maximal body weight of 49 g (determined at euthanasia).

[e]Estimated human consumption based on an average of 3.7 L/person/day as reported by the Institute of Medicine, 2005[68] and an average maximal body weight of 87 kg as reported by the Australian Bureau of Statistics, 2022[69].

[f]BAF = bioaccumulation factor (mean PFAS concentration/water sample).

[g]The PFAS was only detected in 2 of 5 biological replicates.

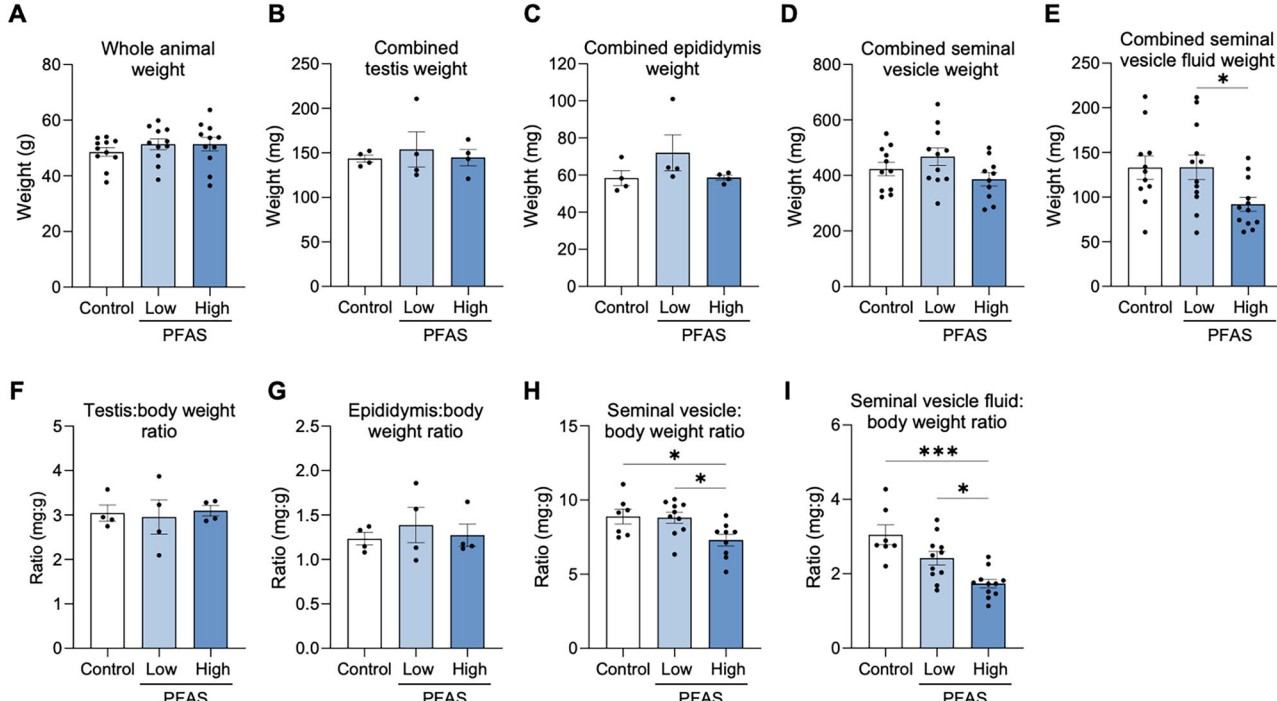

**Fig. 2 | Assessment of the impact of per- and polyfluoroalkyl substances (PFAS) on whole animal and reproductive tissue weights.** Following administration of a 12-week PFAS exposure regimen, mice were culled and **A** whole animal weight was recorded. Upon dissection, the weights of the combined **B** testes, **C** epididymides, **D** seminal vesicles, and **E** seminal vesicle fluid were recorded. These values were used to calculate the ratio of **F** testes, **G** epididymis, **H** seminal vesicles, and **I** seminal vesicle fluid relative to that of the whole body. Data are presented as mean ± SEM, calculated on the basis of $n = 4$ (testis and epididymis weights and whole body weight ratios) or $n = 12$ (whole body weight, seminal vesicle and seminal vesicle fluid weights and whole body weight ratios) biological replicates per exposure group. Data were subjected to one-way ANOVA with Tukey's multiple comparisons test. $*p < 0.05$, $**p < 0.01$.

## Circulating mouse steroid hormones are altered following PFAS exposure

Informed by the negative results of the cellular assays described above, we next examined whether reduced DSP (Fig. 3A) was linked to PFAS-induced endocrine disruption. For this purpose, blood plasma from control and PFAS-exposed mice was analyzed to quantify circulating levels of a panel of steroid hormones (Fig. 4) with a focus on those responsible for influencing sex characteristics. The synthesis pathway and key enzymes responsible for catalyzing the interconversion of these steroids are presented in Supplementary Fig. 1. Among the 15 steroids analyzed, 14 were identified in concentrations above the limit of detection, and of these, the majority were detected at comparable concentrations in the blood plasma of mice across all exposure cohorts. Notable exceptions, however, included the key androgens, testosterone and DHT, the concentrations of which were significantly reduced in mice administered the low dose, but not the high dose, PFAS cocktail ($p < 0.05$; Fig. 4G, H). Mice in the low-dose PFAS group generally presented with lower concentrations of most of the other examined hormones compared with their control or high-dose PFAS counterparts, although differences were not statistically significant. Additionally, corticosterone, the major murine stress hormone, was detected at significantly reduced concentrations in the plasma of the high dose, but not low dose, PFAS-exposed mice ($p < 0.05$; Fig. 4N).

In seeking to account for the reduction in serum testosterone and DHT in mice receiving a low-dose PFAS cocktail, total levels of cholesterol were examined in both the blood plasma and testicular tissue (Fig. 4O, P) of PFAS-exposed animals. Similarly, steroidogenic gene expression (see Supplementary Fig. 1 for an overview of the hormone synthesis pathway and key enzymes responsible for catalyzing the interconversion of these steroids) in the testicular tissue of PFAS-exposed mice was also examined by real-time quantitative polymerase chain reaction (RT-qPCR; Supplementary Fig. 2). The results of the former analysis demonstrated that overall blood plasma cholesterol levels were not influenced by PFAS exposure. However, we did

record a notable reduction in cholesterol levels within the testicular tissue of PFAS-exposed animals; with this reduction reaching statistical significance in response to the high-dose PFAS exposure regimen ($p = 0.0026$). By contrast, among the panel of eight steroidogenic genes analyzed by RT-qPCR, only the *Lhcgr* gene (which encodes the luteinizing hormone/choriogonadotropin receptor) was significantly reduced in the cohort of mice receiving the high dose PFAS exposure ($p = 0.01$; Supplementary Fig. 2A). Notably, the testicular expression of *Srd5a2* and *Hsd17b6* genes, which specify enzymes responsible for interconversion of testosterone and DHT, were not significantly different among any of the treatment groups (Supplementary Fig. 2G, H).

## PFAS exposure does not alter sperm viability nor a range of functional parameters

The demonstrable dysregulation of a subset of dominant androgenic sex hormones in PFAS-exposed mice prompted examination of sperm quality and function. Accordingly, mature spermatozoa isolated from the cauda epididymides of mice underwent functional assessment (Figs. 5–7), confirming that neither sperm viability (Fig. 5A) nor motility (Fig. 5B, C; total and progressive) was impaired by PFAS exposure. Similarly, more nuanced assessment of sperm motility kinematics using computer assisted sperm analysis (CASA) did not detect any PFAS-induced changes (Supplementary Fig. 3) nor did the imposed exposure regimen elicit an increase in mitochondrial reactive oxygen species (ROS) generation (Fig. 5D) or loss of DNA integrity (Fig. 5E, F) within the spermatozoa of PFAS-challenged males.

Additional scrutiny of the functional competence of PFAS-exposed spermatozoa focused on determining their ability to capacitate (Fig. 6A–F), adhere to zona pellucidae (Fig. 6G), fertilize oocytes in vitro, and ultimately support preimplantation embryonic development (Fig. 7). Firstly, the proportion of spermatozoa capable of responding to capacitation stimuli was assessed by quantitating global tyrosine phosphorylation (a key hallmark of capacitation) by immunoblotting (Fig. 6A–D and Supplementary

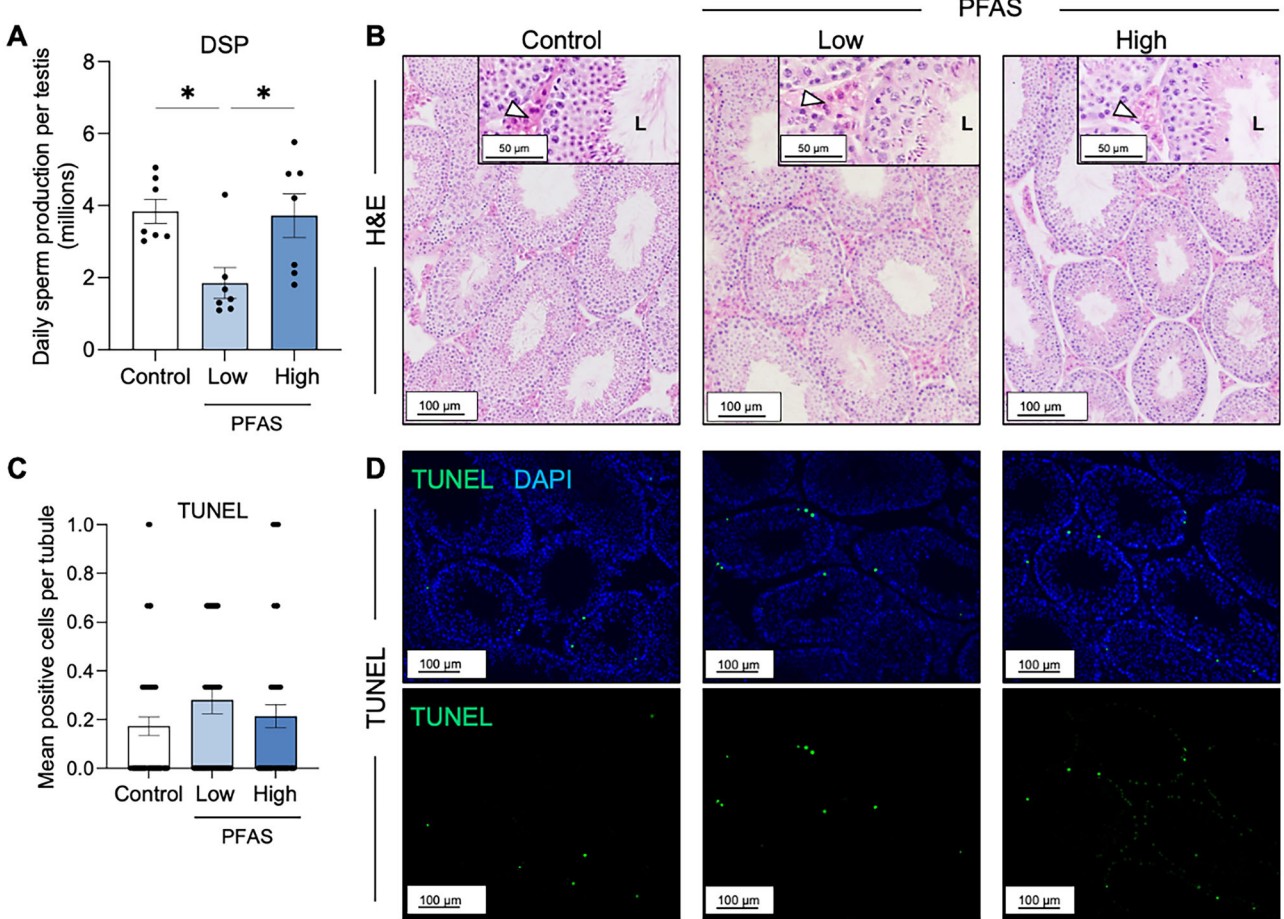

**Fig. 3 | Assessment of the impact of per- and polyfluoroalkyl substances (PFAS) on testicular function. A** Testis function was assessed via calculation of rates of daily sperm production (DSP) using protocols optimized for frozen testis tissue. **B** Gross testis histology was assessed using hematoxylin and eosin (H&E) staining and **C**, **D** a terminal deoxynucleotidyl transferase (TdT)- mediated dUTP nick end labeling (TUNEL) DNA damage assay was used as a surrogate to determine the burden of apoptotic cell death. Arrowheads indicate the interstitial tissue in which the Leydig cell population resides, and L indicates the lumen of the seminiferous tubules. Data are presented as mean ± SEM, calculated on the basis of $n = 3$ biological replicates per exposure group for all assays, except for DSP which had $n = 7$. Data were subjected to one-way ANOVA with Tukey's multiple comparisons test. *$p < 0.05$. Scale bars equal 50 μm for image inserts and 100 μm for main images.

Fig. 4) and by immunocytochemistry (Fig. 6E, F). Immunoblotting demonstrated that PFAS exposure altered the level of phosphotyrosine expression elicited by capacitation stimuli such that there was a significant reduction in overall labeling intensity in spermatozoa from the high dose compared to the low dose PFAS cohort ($p = 0.008$; Fig. 6C); yet neither of the PFAS-exposed groups were statistically different from the control. Densitometric analysis of dominant phosphotyrosine labeled bands (labeled A–G in Fig. 6A, B, D) revealed only one band (band D; ~110 kDa) to be differentially labeled between the control and PFAS exposure cohorts (Fig. 6D; control vs low PFAS $p = 0.02$). Nevertheless, PFAS exposure did not influence the distribution of phosphotyrosine labeling nor the proportion of labeled spermatozoa as analyzed by immunocytochemistry (Fig. 6E, F). Similarly, PFAS exposure also failed to significantly impact the ability of spermatozoa to recognize and adhere to the zona pellucidae of oocytes in an in vitro setting (Fig. 6G). Further extending these functional analyses, IVF and embryo culture experiments were conducted to determine if PFAS exposure influenced downstream fertilization and/or embryo development (Fig. 7). These assays revealed that PFAS-exposed sperm retained the ability to fertilize oocytes (Fig. 7A) and to support embryo development from the 2-cell stage (Fig. 7B) through to the blastocyst stage (Fig. 7C) at rates that were indistinguishable from those of their unexposed control counterparts. Moreover, the timing of progression through landmark stages of early embryonic development was also unchanged among embryos generated from PFAS-exposed spermatozoa (Fig. 7D–G).

## The small non-coding RNA profile of spermatozoa is responsive to PFAS exposure

Based on a growing body of data that indicates males are capable of responding to environmental exposures by relaying epigenetic stress signatures to their spermatozoa[35,36], the sncRNA landscape of PFAS-exposed sperm was assessed (Fig. 8). This analysis revealed the sperm sncRNA profile was indeed responsive to PFAS, with proportional increases in rRNA fragments (also known as rsRNAs) and microRNAs (miRNAs) in sperm populations from PFAS exposed males (Fig. 8A). Conversely, the contribution of the tRNA derived RNA fragments (tRFs; also referred to as tRNA derived RNAs (tDRs) or as tsRNAs) to the global sncRNA population was decreased in PFAS exposed sperm (Fig. 8A and Supplementary Table 2). This trend was primarily attributed to a significant reduction in the proportion of tRFs of 31 and 32 nucleotides (nt), in the spermatozoa of the low ($p = 0.003$; Fig. 8B and Supplementary Data 1) and high PFAS exposure groups ($p = 0.04$; Fig. 8B and Supplementary Data 2). At the level of individual tRFs, 12 (2.8%) were significantly increased in the spermatozoa of males exposed to the low PFAS cocktail, whereas 16 (3.7%) were decreased and only one increased (0.2%) in the high PFAS exposure group compared to unexposed control spermatozoa (Fig. 8E). Notably, no sperm tRFs displayed similar significant changes in abundance in both PFAS exposure cohorts. In contrast to the altered profile of tRFs, the relative proportion of miRNAs was only subtly increased in the spermatozoa of high PFAS-exposed males (Fig. 8A); a trend that held across miRNAs of between 20 and 23 nt, albeit non-significantly

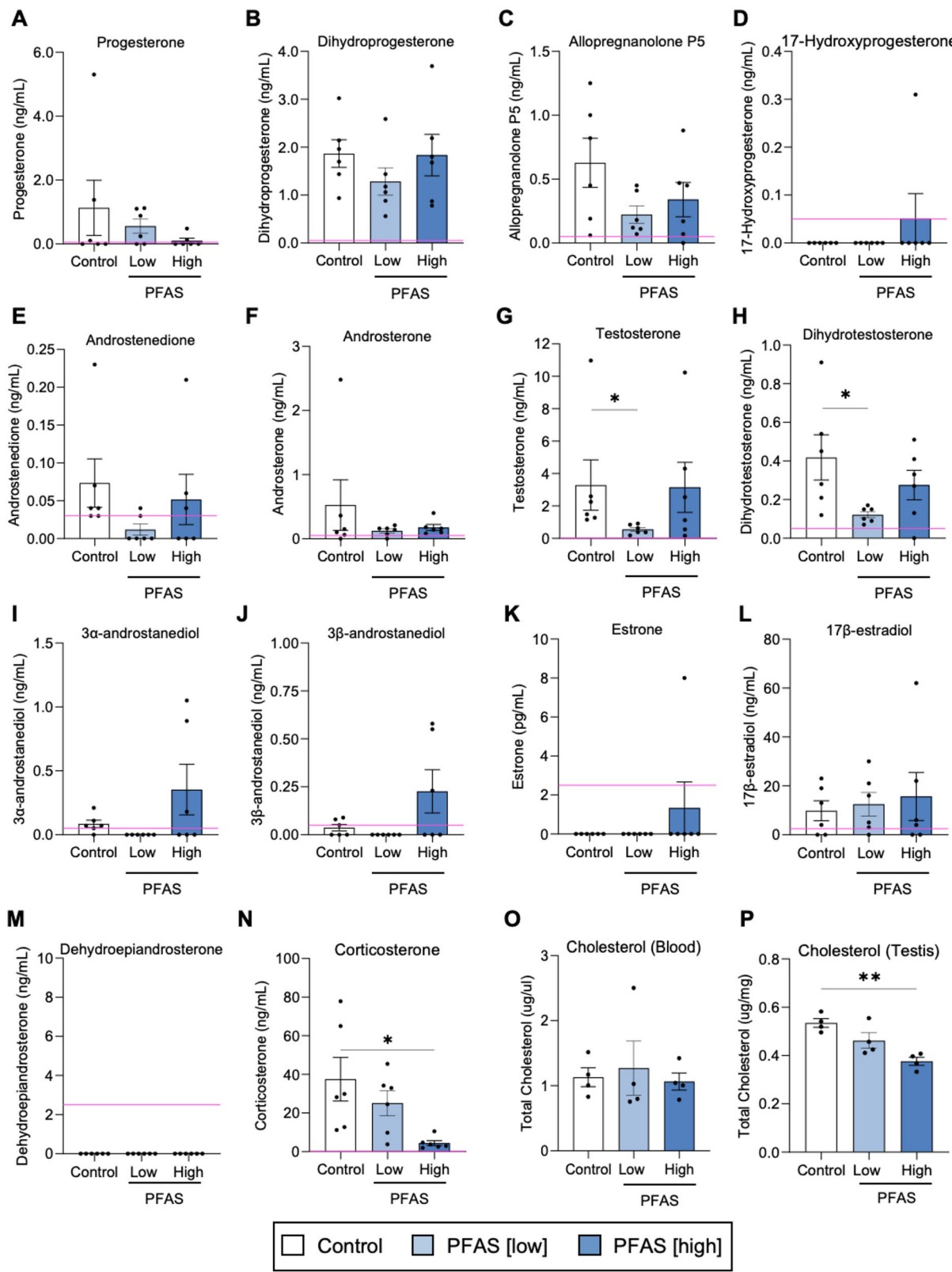

(Fig. 8C). The most profound abundance changes in individual miRNAs were recorded in the spermatozoa of high PFAS-exposed males in which some 113 miRNAs were significantly altered, with three (0.3% of all identified miRNAs) decreased and 110 (10.4%) increased in abundance compared with control spermatozoa (Fig. 8F; Supplementary Data 2). By comparison, this response was far more subtle in the spermatozoa of low PFAS-exposed males in which only 25 miRNAs (2.3%) were shown to display altered abundance

(Fig. 8F; Supplementary Data 1). Among the subset of dysregulated miRNAs, four (*miR*-203, *miR*-3620, *miR*-7682, *miR*-6966) were consistently altered in the spermatozoa of both the low and high PFAS groups (Fig. 8F; green dots). Expanding this analysis to consider PIWI-interacting RNAs (piRNAs) revealed modest impacts of PFAS on the overall abundance of this sncRNA species (Fig. 8A), with no changes in the distribution of nucleotide length across the exposure groups (Fig. 8D). More specifically, only one (0.2%)

**Fig. 4 | Assessment of the impact of per- and polyfluoroalkyl substances (PFAS) on circulating steroid hormone levels.** Blood plasma was assessed for a panel of steroidal hormones by an independent National Association of Testing Authorities (NATA) accredited laboratory. The relative abundance of 14 of these hormones is shown. **A** Progesterone, **B** Dihydroprogesterone, **C** Allopregnanolone, **D** 17-Hydroxyprogesterone, **E** Androstenedione, **F** Androsterone, **G** Testosterone, **H** Dihydrotestosterone, **I** 3α-Androstanediol, **J** 3β-Androstanediol, **K** Estrone, **L** 17β-estradiol, and **M** Dehydroepiandrosterone. In addition to the panel of reproductive hormones, the levels of the major stress hormone, corticosterone (**N**) and total cholesterol levels (**O, P**) were also assessed in each exposure group. All graphical data are presented as mean ± SEM, calculated on the basis of $n = 2–6$ biological replicates per exposure group. Pink lines indicate the limit of detection for each hormone. Hormones present above the limit of detection in more than 2 samples for each exposure group were subjected to one-way ANOVA with Tukey's multiple comparisons test (if normally distributed) or Kruskal–Wallis test (if not normally distributed). *$p < 0.05$.

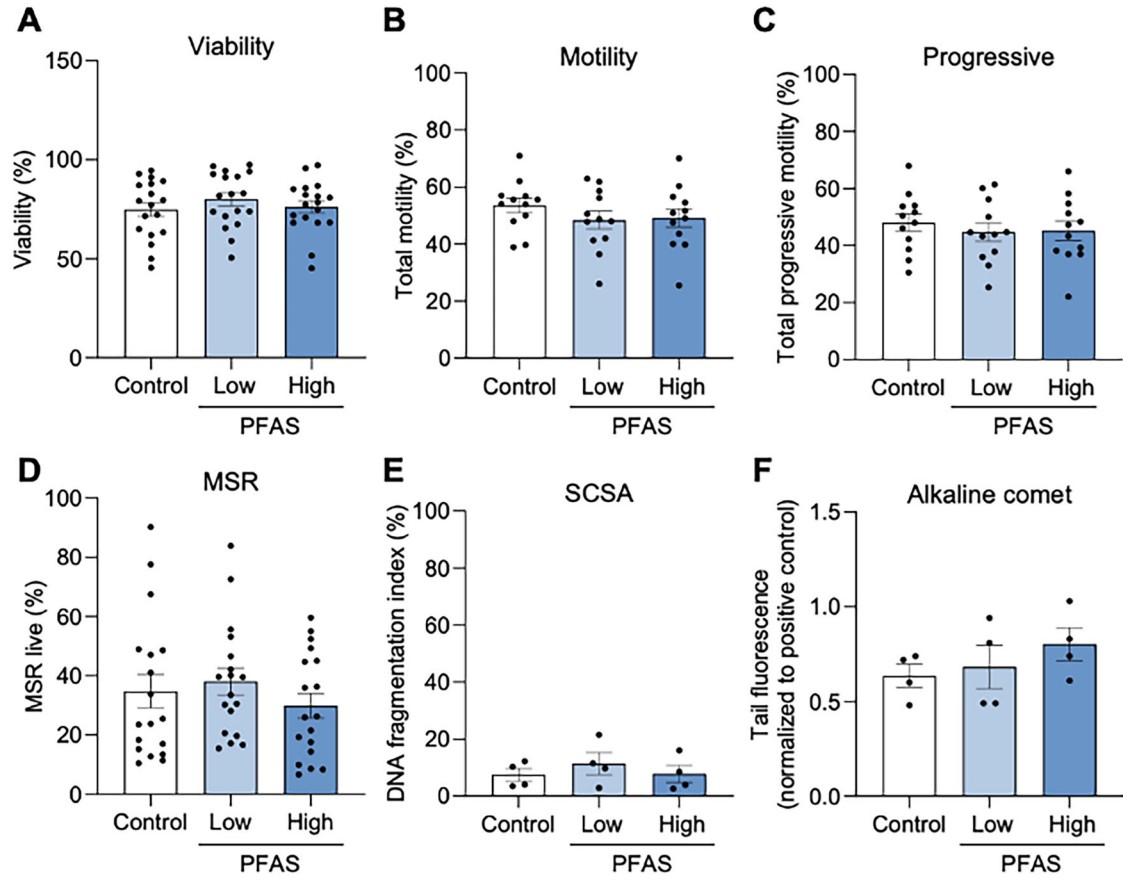

**Fig. 5 | Assessment of the effect of in vivo per- and polyfluoroalkyl substances (PFAS) exposure on sperm cell integrity.** Mature spermatozoa were isolated from the cauda epididymides of mice in each exposure group before being immediately assessed for **A** viability using a Sytox Green viability stain, and **B** total and **C** progressive motility using computer assisted sperm analysis (CASA). Alternatively, the cells were subjected to a **D** MitoSOX Red (MSR) assay to determine the level of mitochondrial reactive oxygen species (ROS) generation. Genome integrity was assessed using a combination of the **E** sperm chromatin structure assay (SCSA) to determine DNA fragmentation and **F** an alkaline Comet assay to detect DNA strand breakage. Data are presented as mean ± SEM, calculated on the basis of $n = 12$ biological replicates for motility, $n = 18$ for viability and MSR, and $n = 4$ biological replicates for SCSA, and Comet. All data were subjected to one-way ANOVA with Tukey's multiple comparisons test.

piRNA was decreased in abundance and a further 31 (6.6%) increased in abundance in the spermatozoa of low PFAS-exposed males (Fig. 8G; Supplementary Data 1) and this was refined to only a single piRNA (0.2%) that displayed reduced abundance in the high PFAS-exposed spermatozoa (Fig. 8G; Supplementary Data 2) but was conserved across both PFAS exposures (Fig. 8G; green dot). This analysis also revealed that exposure to PFAS resulted in the modified abundance of several mRNA fragments (59 and 156 differentially abundant in the low and high PFAS group, respectively) but failed to elicit any such changes in rRNA fragments compared to control spermatozoa (Supplementary Fig. 5).

### Preimplantation gene expression is dysregulated in embryos fertilized by spermatozoa of PFAS-exposed males

In consideration of the potential for altered sperm sncRNA profiles (Fig. 8) to influence patterns of early embryonic gene expression, 4-cell embryos generated via IVF with oocytes retrieved from unexposed females and the spermatozoa of PFAS-exposed males were subjected to transcriptomic analyses (Fig. 9). Selection of this embryonic stage was timed to follow the robust wave of zygotic genome activation that occurs at 2-cell stage in mouse embryos and in line with literature indicating that paternal sncRNA changes affect embryo gene expression at this stage of their development[35]. This strategy revealed broadly equivalent impacts on embryonic gene expression resulting from paternal PFAS exposure regardless of dose. Specifically, 118 (1.6%) differentially expressed genes (DEGs; $p ≤ 0.05$ and fold change ±1.5) were identified as being down-regulated, and an equivalent number up-regulated (118; 1.6%) in embryos from the low PFAS exposure group (Fig. 9A; Supplementary Data 3). In the high PFAS-exposed group, 168 (2.3%) and 111 (1.5%) DEGs were down- and up-regulated compared with control embryos (Fig. 9B; Supplementary Data 4). Modest conservation was evident within the up- and down-regulated embryonic DEGs across both PFAS exposure groups (i.e., 67 conserved DEGs, representing ~29% of low PFAS and 25% of high PFAS DEGs; Fig. 9A, B; green dots). The strength of

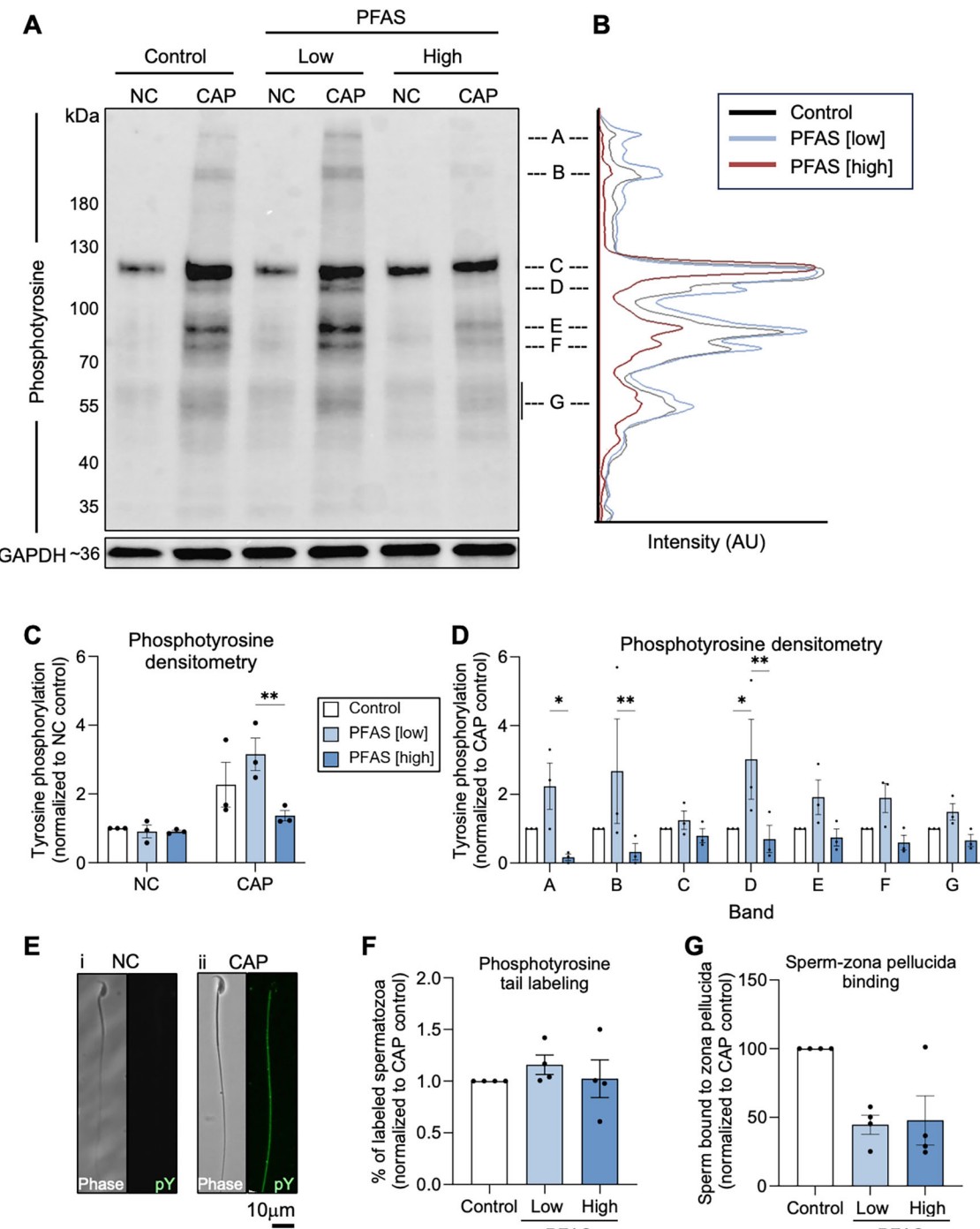

**Fig. 6 | Assessment of the effect of per- and polyfluoroalkyl substances (PFAS) exposure on sperm capacitation.** Cauda epididymal spermatozoa isolated from both PFAS-exposed and control mice were assessed for their ability to respond to capacitation-inducing stimuli. Immunoblotting was used to detect relative levels of protein phosphotyrosine (pY) expression. **A** A representative immunoblot and **B** corresponding densiometric trace of phosphotyrosine band intensity and graphical depictions of labeling intensity as measured **C** across the entire sample lane or **D** specifically across each of seven most prominent phosphotyrosine bands. These studies were complemented by the use of immunofluorescence to determine **E** the localization of phosphotyrosine labeling (predominantly in the sperm flagellum) and **F** the percentage of labeled spermatozoa. **G** Interrogation of the functional capacity of PFAS-exposed spermatozoa featured an assessment of their ability to bind to the zona pellucidae of mouse oocytes. All graphical data are presented as mean ± SEM, having been calculated on the basis of $n = 3$ biological replicates per exposure group for densitometry and $n = 4$ for phosphotyrosine tail labeling and sperm-zona pellucida binding. Data were subjected to one-way ANOVA with Tukey's multiple comparisons test. *$p < 0.05$. Acronyms referred to include NC non-cap, CAP capacitated, kDa kilodaltons, AU arbitrary units. Scale bars equal 10 μm.

the association in embryonic gene dysregulation between these exposure groups was supported by a Pearson's correlation coefficient ($r$) of 0.808 (Supplementary Fig. 6A). Conceivably, the alterations in the sperm sncRNA profile could lead to the documented gene dysregulation. To investigate this, we identified the gene targets of the altered miRNAs in low and high-PFAS

spermatozoa using Target Scan software[37,38]. This investigation identified 12 and 24 genes that were differentially expressed in low and high-PFAS fertilized embryos, respectively, and were predicted targets of the altered miRNAs (Supplementary Data 3 and 4). Four of these predicted gene targets were conserved across both PFAS exposure groups ADP-ribosylation factor

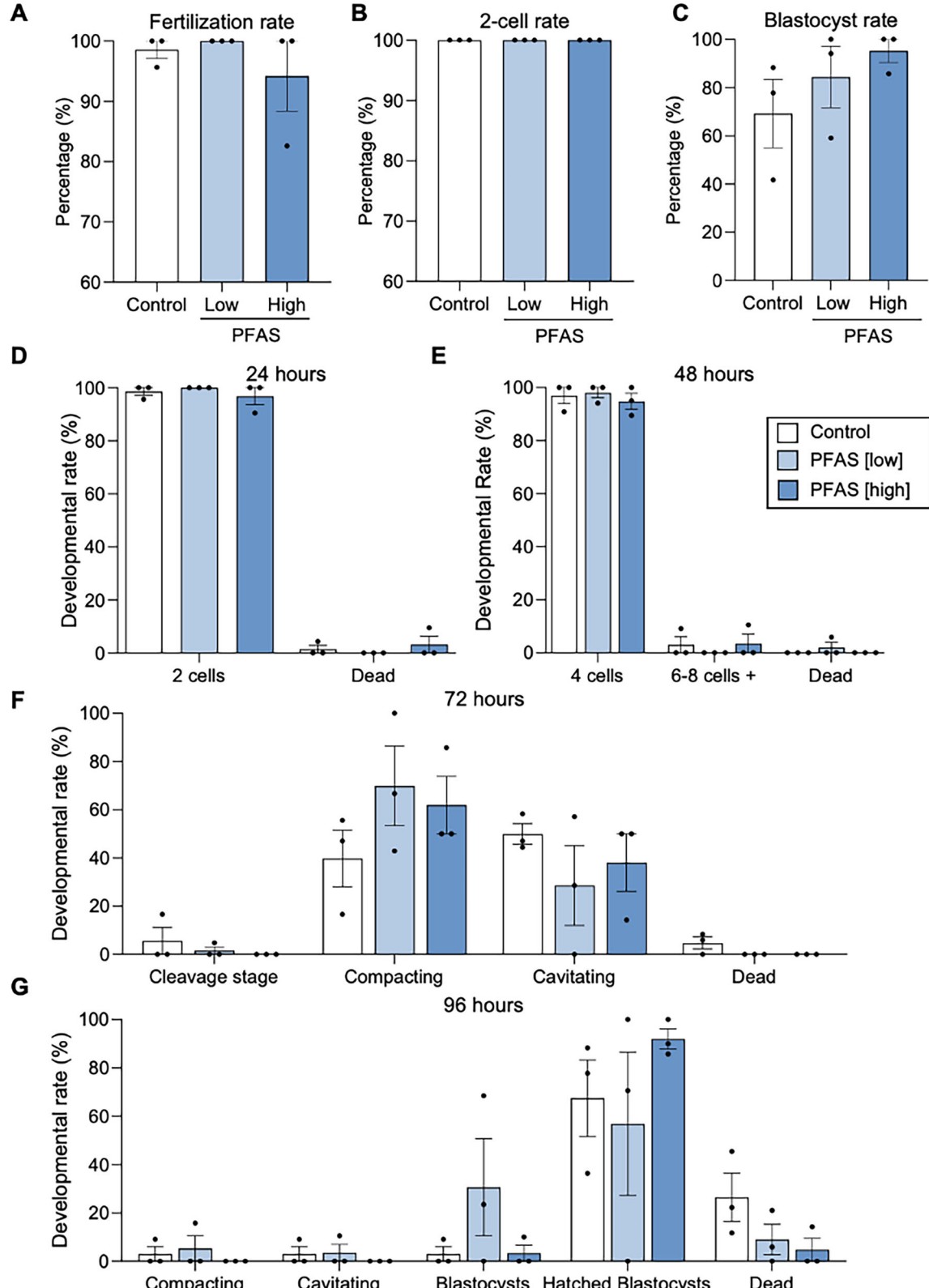

**Fig. 7 | The impact of paternal per- and polyfluoroalkyl substances (PFAS) exposure on the ability of spermatozoa to fertilize and support early embryonic development.** Spermatozoa from both PFAS exposed and unexposed males were assessed for their ability to fertilize and support pre-implantation embryonic development via in vitro fertilization (IVF) and embryo culture. **A** Fertilization rate was recorded as the percentage of zygotes relative to the total number of MII oocytes co-incubated with spermatozoa. **B** The proportion of embryos progressing to 2-cell and **C** blastocysts was recorded relative to the total number of zygotes produced. Embryo development was tracked for 96 hours (h) with the rates of development being recorded at **D** 24, **E** 48, **F** 72, and **G** 96 h after fertilization. All graphical data are shown as mean ± SEM while statistics were carried out on arcsine transformed data to account for the use of proportional data. A minimum of 19 oocytes were subjected to IVF from each exposure group across three biological replicates (i.e., 3 male mice per exposure group) equating to a total of 71, 80, 72 oocytes from the control, low PFAS and high PFAS groups, respectively. Differences between groups were assessed with **A–C** one-way ANOVA or **D–G** 2-way ANOVA, with Tukey's multiple comparison test.

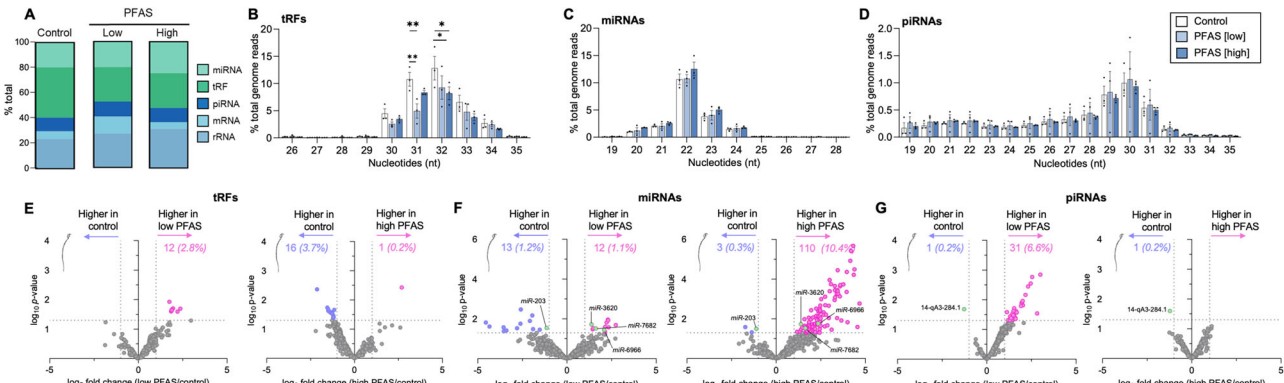

**Fig. 8 | The impact of per- and polyfluoroalkyl substances (PFAS) exposure on the sperm small non-coding RNA (sncRNA) profile.** SncRNAs (18–40 nucleotides) were sequenced from control, low PFAS and high PFAS-exposed populations of mouse spermatozoa. **A** The average contribution of each sncRNA species to the overall sperm sncRNA profile is depicted, as is the size (nt; nucleotide) distribution of sncRNAs mapped to **B** tRNA fragments (tRFs; also referred to as tRNA derived RNAs (tDRs) or as tsRNAs) and **C** microRNA (miRNA) and **D** piwi-interacting (piRNA) subclasses. Data are presented as a percentage of total genome reads for sequences of each length. Volcano plots are included to illustrate the $\log_2$ fold change and $\log_{10}$ p value for **E** tRFs, **F** miRNAs, and **G** piRNAs identified in the spermatozoa of males exposed to low PFAS (left) and high PFAS (right). Colored dots are used to indicate differentially accumulated sncRNAs as determined by DESeq2. Purple is used to represent reduced abundance in PFAS-exposed spermatozoa, and pink indicates an increased abundance. Green highlights those miRNAs with altered abundance in both the low and high PFAS groups. The applied cutoff criteria for altered sncRNAs was a fold-change ≥1.5 and $p \le 0.05$. All analyses were based on $n = 3$–4 male mice per exposure group. mRNA refers to messenger RNA and rRNA refers to ribosomal RNA.

3 (*Arf3*), Podoplanin (*Pdpn*), SID1 transmembrane family member 2 (*Sidt2*), and Ras-related protein Rab-39 (*Rab39*).

To assess the potential implications of PFAS-induced changes in embryonic gene expression, DEGs were analyzed by Ingenuity Pathway Analysis (IPA) software to identify the subset of canonical pathways, upstream regulators (Supplementary Fig. 6B–E), and diseases they putatively influence (Fig. 9C, D). Key to this analysis, the predicted extent of dysregulation is denoted by a p value of ≤0.05 and a Z-score, whereby anything above a threshold of ≥2 is considered "activated", while conversely, anything ≤−2 is predicted to be "inhibited". Among the 9 most highly dysregulated diseases to which low PFAS responsive DEGs were mapped, only "size of body" reached the threshold to be considered activated, while eight diseases were inhibited including "death of embryo", "growth failure", "growth failure or short stature", and "abnormal metabolism" (Fig. 9C). In the embryos generated from high PFAS-exposed spermatozoa, IPA curated four dysregulated diseases ($p \le 0.05$; Z-score ≥ 2 or ≤ −2; Fig. 9D), of which two were activated (i.e., "development of hematopoietic system" and "development of bone marrow") and two were inhibited; "cell death of blood cells" and "synthesis of carbohydrate". Importantly, several DEGs highlighted by this IPA analysis that were associated with diseases such as "size of body" and "development of hematopoietic system" were revealed to be targets of the PFAS responsive sperm miRNAs (Fig. 9E, F).

In extending this in silico analysis to identification of the canonical signaling pathways that may contribute to these disease states, we identified 15 dysregulated pathways ($p \le 0.05$; Z-score ≥ 2 or ≤ −2) in embryos generated from the sperm of low PFAS-exposed males (Supplementary Fig. 6B). Those pathways which met the threshold requirement that is considered activated included several tumor suppression pathways (e.g., "p53 signaling", "endocannabinoid cancer inhibition pathway", while those pathways predicted to be inhibited included "ERK/MAPK signaling", "PDGF signaling", and "HGF signaling" (Supplementary Fig. 6B). Similarly, 11 pathways were dysregulated in the embryos fertilized by spermatozoa from high PFAS-exposed males (Supplementary Fig. 6C), with those considered activated including "deubiquitination" and pathways relating to "D-myo-inositol biosynthesis". The opposing inhibited pathways in the high PFAS group included "telomere maintenance", "DNA replication pre-initiation", "serotonin receptor signaling", and "processing of capped intron-containing pre-mRNA".

The IPA upstream regulator prediction tool was subsequently employed to identify the upstream regulators of these altered pathways in the 4-cell embryos (Supplementary Fig. 6D, E). This analysis revealed 18

predicted ($p \le 0.05$; Z-score ≥ 2 or ≤ −2) upstream regulators in the low PFAS exposure group (Supplementary Fig. 6D), while the high PFAS exposure group indicated 14 predicted upstream regulators (Supplementary Fig. 6E). Among these predicted upstream regulators, there was an overlap of one highly predicted across both PFAS exposure groups, namely Nuclear protein, transcriptional regulator 1 (NUPR1).

## Discussion

In our previous work, we assessed the direct impact of an environmentally relevant cocktail of PFAS on mouse spermatozoa using an in vitro exposure regimen[29]. Together with the inherent limitations imposed by a short-term in vitro exposure protocol, the findings of this previous work prompted the current study, in which we assessed the in vivo reproductive implications of this same environmentally relevant cocktail of PFAS under conditions that simulate human exposure. The major findings of this study demonstrate that, despite significant bioaccumulation in the blood plasma and testes of exposed animals, consumption of the PFAS cocktail did not elicit overt cytotoxicity or genotoxicity within either the somatic cells of the male reproductive tract or that of the developing male germline, as would be expected given the existing knowledge surrounding human exposures[23,24,39]. Notwithstanding these results, PFAS exposure did reduce the circulating concentration of androgenic sex steroids and testicular cholesterol levels, which correlated with reductions in the rate of DSP and the weight of seminal vesicle fluid relative to that of overall body weight. Additionally, the mature spermatozoa of exposed males also presented with an altered sncRNA profile that appeared to instigate aberrant gene expression at the 4-cell embryo stage. Collectively, these data indicate that paternal PFAS exposure may have significant consequences beyond that of the exposed father to include the development and health of their offspring.

Our collective data support previous reports identifying the testis as a vulnerable organ for PFAS-mediated effects, possibly reflecting its propensity to sequester PFAS[13,16,18]. Indeed, we recorded significant PFAS bioaccumulation in the testes (and blood plasma) of exposed animals. Despite this, the administered PFAS cocktail did not elicit overt cytotoxicity or genotoxicity within the cells of the male reproductive tract tissues/organs examined (i.e., testes and epididymis) nor the mature spermatozoa. The low (i.e., environmentally relevant) PFAS cocktail did, however, significantly compromise daily rates of sperm production, a finding that mirrors epidemiological evidence of reduced sperm production in a Danish population of men with elevated serum PFOA and PFOS levels[24]; two PFAS also included in our study cocktail.

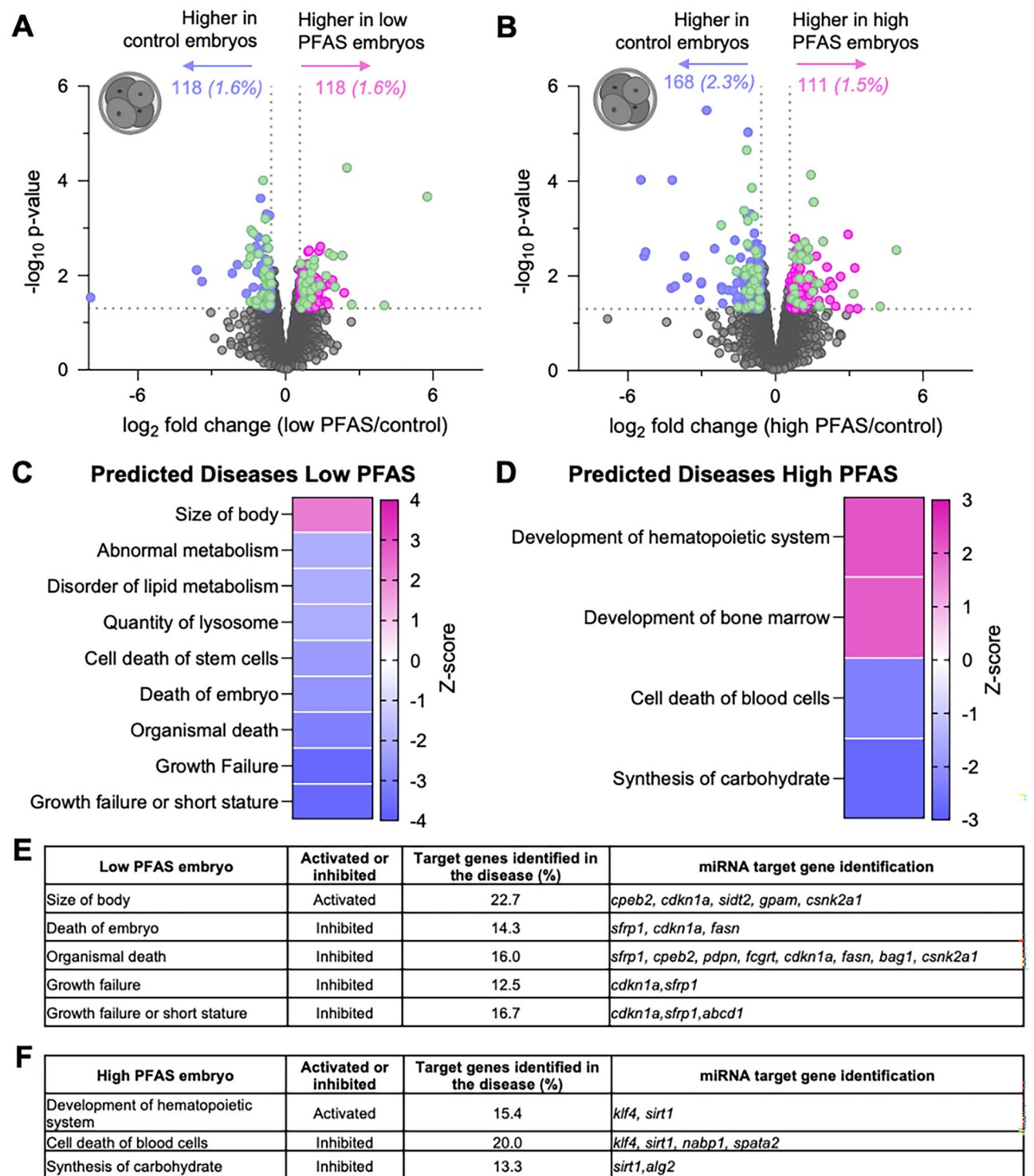

**Fig. 9 | The impact of paternal per- and polyfluoroalkyl substances (PFAS) exposure on embryonic gene expression in 4-cell embryos.** Volcano plots illustrate the gene expression profiles in 4-cell embryos fertilized by control spermatozoa or sperm from male mice exposed to **A** low and **B** high PFAS. Genes with $p \leq 0.05$ and fold-changes ±1.5 were considered to have differential gene expression (DEGs) and are denoted by either pink or purple dots to identify those as being significantly up- and down-regulated. Alternatively, green dots were used to denote those DEGs with expression that changed consistently across both exposure regimens. Interrogation by ingenuity pathway analysis (IPA) software was used to identify the diseases predicted to be affected by embryonic DEGs ($p \leq 0.05$) in the **C** low and **D** high PFAS groups. The displayed heatmaps are based on a $p < 0.05$ and a Z-score scale where a score of ≥2 is considered "activated" and a score ≤ −2 is considered "inhibited". Tables showing the genes targeted by altered sperm microRNAs (miRNAs) in the **E** low and **F** high PFAS groups. Created in BioRender. Nixon, B. (2025) https://BioRender.com/eq3tqoz (Agreement number: QT28OPEW0P).

In seeking to reconcile the mechanistic basis of such responses, PFAS exposure was shown to reduce the circulating concentration of the dominant androgenic sex steroids, testosterone and DHT, as well as the major murine stress hormone, corticosterone. These effects exhibited a non-monotonic (i.e., nonlinear) response such that testosterone and DHT concentrations were only significantly inhibited in mice exposed to the environmentally relevant concentration (i.e., the low PFAS dose). This trend held across most of the hormones examined, whereby the greatest impact on

circulating concentrations was detected in the low-dose PFAS group compared with their control or high-dose PFAS counterparts. Such findings are not without precedent, with numerous experimental studies having identified unconventional dose-response relationships among endocrine-disrupting chemicals[40,41], including PFAS. By way of example, it has been reported that pathologies such as cognitive function[42] and the prevalence of diabetes[43,44] exhibit a non-monotonic relationship with plasma PFAS concentrations. Accordingly, several molecular mechanisms have been posited to explain such phenomena, including receptor desensitization, negative feedback with increasing dose, and/or dose-dependent metabolism modulation[41].

These data are further supported by our BAF calculations, which indicate that the mechanisms of action may differ even between PFAS species. As an example, PFHxS and PFHpS had calculated plasma BAF values approximately 2–3 fold higher than the more hydrophobic PFOS (i.e., 28.19 and 20.47 vs 11.29, respectively). Such differences likely indicate that these PFAS species have different binding sites in vivo, since the degree of hydrophobicity influences how readily a PFAS is taken up by an organism[45]. Thus, a higher hydrophobicity lends itself toward greater PFAS uptake[45]. By contrast, in the situation described above, the more hydrophobic PFOS had the lower BAF value (i.e., less had been taken up from the drinking water by the mouse). Notably, the accumulation propensity of each PFAS investigated in this study appeared to scale proportionally with that of the initial administered concentration, such that broadly equivalent BAF values were recorded between the exposure groups. By way of example, for the sum total PFAS, the blood plasma of animals exposed to low and high PFAS had calculated BAF values of 12.48 and 12.45, respectively, while for the testis, the calculated BAF values were 3.40 and 3.57 in these cohorts. Such comparable BAF values among the two exposure groups suggest that the PFAS doses administered in this study did not exceed the maximum uptake thresholds in either the plasma or testis of exposed animals; a potentially troubling scenario when considered in the context of human exposure.

Our data add to a growing body of literature linking PFAS, and in particular PFOA and PFOS, to endocrine-disrupting activity. Indeed, independent studies conducted in rat models have identified strong correlations between PFOS exposure and reduced serum testosterone after short-term 14-day (50 mg/kg)[46] and 28-day (1 and 3 mg/kg/day) exposure regimens[47]. Similarly, studies in mice have documented reduced serum testosterone following 10 mg/kg/day PFOS exposure for 21[48] and 35-days[49]. Reminiscent of our own findings, such endocrine changes were accompanied by a significant reduction in sperm count[49]. They were, however, also associated with increased apoptosis in germ cells[49] and histopathological changes in the anterior pituitary, hypothalamus and testis; presenting as edema surrounding the seminiferous tubules and malformation and degeneration of spermatid heads[47]. Such pronounced pathological responses were not observed among the reproductive tissue assessed in our own study. This is likely due to the aforementioned studies using much higher dosages of PFOS in comparison to the present study (that is, in our study PFOS was administered at 0.014 (low) and 0.159 (high) mg/kg/day PFOS, whereas the previously mentioned studies featured PFOS at 1–3 mg/kg/day and 10 mg/kg/day). Importantly, the data presented here also agree with human epidemiological evidence that reports increased serum PFAS in adult men [particularly PFOS (4–15 μg/L)[22], and PFOA (1-96 μg/L)[50]], correlates with decreased circulating testosterone. Indeed, in our study, the mice receiving the low dose PFAS had plasma PFOA levels of 80.6 ± 15.28 μg/L, which translates to 0.0012 mg/kg/day for mice and is equivalent to 0.0004 mg/kg/day in an adult human (average weight of 87 kg). Concerningly, a recent factsheet published by the National Health and Medical Research Council (NHMRC) revealed the drinking water quality guideline value should not exceed 0.008 μg/L for PFOS[30], meaning that the doses in this study are 13,750–1,625,000 times those considered safe for humans for the low and high doses, respectively.

Among the putative mechanisms that could account for reduced androgenic sex hormone concentrations, it has been proposed that PFAS induced tissue damage may negatively impact the hypothalamic–pituitary–testicular axis[47] or alternatively, that PFAS directly interfere with the expression and/or activity of key enzymes [including, cytochrome P450 17A1 (*Cyp17a1*), 3β-hydroxysteroid dehydrogenase (*3beta-HSD*), and 17β-hydroxysteroid dehydrogenase (*17beta-HSD*)] that regulate testicular steroid hormone synthesis pathways[46,48]. In exploring the latter mechanism, we failed to document any dysregulation in the expression of eight genes responsible for the expression of key steroid synthesis enzymes (Supplementary Fig. 2). We are cautious to note, however, that gene expression profiles do not necessarily equate to enzyme abundance or activity and thus advocate for further investigation focusing on enzyme activity assessment in addition to quantifying intra-testicular steroid hormone levels. Additionally, whilst not undertaken in this study, to fully elucidate the mechanisms underscoring the endocrine disrupting activity of PFAS, a detailed functional analysis of the hypothalamic–pituitary–testicular axis should be undertaken in parallel with activity measurements of key enzymes and steroid receptors within these tissues.

Notwithstanding reduced sperm production as a key consequence of the imposed PFAS exposure regimen used in this study, the population of mature spermatozoa that were recovered from the storage reservoir of the cauda epididymis displayed no overt signatures of DNA damage and retained normal motility profiles, as might have been expected based on existing human studies[23,24,39]. Although these cells were characterized by subtle changes in their ability to capacitate, this did not compromise overall rates of IVF or progression of the embryos generated through the early stages of preimplantation development. A more detailed interrogation of the molecular impact of PFAS exposure on sperm biology revealed pronounced changes in the overall sncRNA signature of these germ cells; with sncRNAs belonging to the miRNA, tRF, and piRNA species being particularly influenced. Impetus for this analysis stemmed from evidence that the composition of sperm sncRNAs displays considerable plasticity in response to changes in the paternal environment, including sensitivity to factors such as diet[36,51,52], stress[53,54], exercise[55], and heat[56]. Consistent with the results of this study, recent research has also highlighted the sensitivity of the sperm epigenome to alternative environmental toxicants, which not only have the prosperity to disrupt sperm sncRNA profiles but also induce phenotypic transmission to offspring[57,58]. Indeed, the significance of such changes rests with repeated demonstration that sncRNAs act as important modifiers of embryonic development, capable of exerting regulatory control over the first zygotic cell division[59], early gene activation[60,61], and implantation[35]. Moreover, such changes to the sperm sncRNA profile have been irrefutably implicated in altering the trajectory of early fetal development and facilitating the transmission of acquired phenotypes to the offspring[35,62], effectively eliciting transgenerational responses reflective of the periconceptional environment of the father. In keeping with this mode of action, transcriptomic profiling in our study revealed in vivo PFAS exposure induces alterations to miRNA abundance, whereby four were altered in both the low and high PFAS-exposed sperm (e.g., *miR-203, miR-3620, miR-7682*, and *miR-6966*). Interestingly, the gene targets of these miRNAs are involved in diseases including "size of body" and "development of hematopoietic system", with four genes being conserved across the two PFAS exposure groups (*Arf3, Pdpn, Sidt2*, and *Rab39*). Notably, a recognized limitation of the traditional ligation-dependent RNA-seq protocol used herein is that it has reduced sensitivity for the detection of highly modified sncRNAs, such as subsets of tRFs and rRNA fragments, which may therefore be underrepresented in our analysis.

These data present the possibility that PFAS-exposed sperm retain their ability to fertilize and can thereby pass on the legacy of this exposure to their offspring. Indeed, the complementary transcriptomic profiling examining the embryonic gene expression in 4-cell embryos fertilized by these sperm identified the activation of disease functions relating to several cancers, "size of body", "disorder of lipid metabolism", and "growth failure" in the low PFAS group, all of which have been linked to PFAS exposure in human studies[18,63,64]. Based on previous work, we were particularly interested in the size of body changes[62]. These data suggest that paternal PFAS exposure

at environmentally relevant levels (in this case, low PFAS) may alter the size of pups sired by these males. Thus, future in vivo breeding studies with PFAS-exposed males will be important to fully elucidate the long-term impact of paternal exposures on offspring health and development. Such future studies would also benefit from taking into consideration that fathers are unlikely to be exposed in isolation, and thus offspring may also be prone to the cumulative effects of PFAS resulting from maternal and/or in utero exposures. This troubling scenario stresses the need for additional high-quality research into the transgenerational impacts of PFAS exposure.

## Conclusions

This study is one of the first to use an environmentally relevant PFAS profile and concentrations found at a recognized contamination site[28]. Collectively, the data presented here identify that exposure of adult male mice to this PFAS mixture has negative consequences for their reproductive function, which manifests in reduced levels of circulating testosterone and DHT, accompanied by a significant, parallel decrease in the rate of DSP. Along with our evidence of significant PFAS bioaccumulation within testicular tissue homogenates, these data affirm the male reproductive tract as a vulnerable site of PFAS action. However, given that mice were only exposed for 12 weeks, we cannot comment on the impact of longer-term PFAS exposure. Nevertheless, the fact the spermatozoa of PFAS-exposed males retained the ability to achieve fertilization, at least in an in vitro context, raises the concerning prospect that these cells may pass on the legacy of PFAS exposure to the embryo upon fertilization. This model is given credence by the demonstration that PFAS-exposed spermatozoa harbor an altered epigenome and produce embryos bearing the hallmarks of dysregulated gene expression linked to tumorigenesis, and aberrant offspring phenotypes such as dysregulation of "size of body". These data imply that offspring born from male mice previously exposed to PFAS may experience the effects of this exposure without having been directly exposed themselves. However, we caution that further studies and additional phenotypic assessments are necessary to validate the functional consequences of the observed molecular changes in sperm and embryos before concluding the potential for perpetuation of transgenerational or multigenerational effects among the offspring of PFAS-exposed individuals.

## Material and methods
### Chemical reagents

All reagents and chemicals used in this study were of research grade and were purchased from Merck (Merck Group; Darmstadt, Germany) unless stated otherwise. For all experiments involving spermatozoa, cells were diluted and incubated in freshly prepared Biggers, Whitten, and Whittingham (BWW) medium (osmolarity between 290 and 310 mOsm/kg)[65] pre-warmed to 37 °C.

### Formulation of PFAS cocktail

The composition of the PFAS cocktail used in this study was formulated as previously described[29] to emulate the concentration and composition of the PFAS profile identified within a contaminated water well located at Williamtown, NSW, Australia[28]. This site was designated a PFAS "Red-Zone"[66] owing to contamination arising from the use, storage, and disposal of fire-fighting foams on an adjacent Royal Australian Air Force Base[19]. Specifically, the PFAS cocktail comprised nine PFAS (Table 1) detected in a representative Williamtown well[28]. This included PFOS, PFHxS and PFOA, which represent three of the most commonly identified PFAS in serum biomonitoring programs[67]. This cocktail was administered to mice via their drinking water at either equivalent levels or at levels 10-fold higher than those identified at Williamtown; referred to in this study as low and high dose PFAS exposures. The latter high dose exposure regimen was implemented to approximate the experience of occupationally exposed, as opposed to residentially exposed, individuals[13,14]. Prior to experimentation, drinking water was analyzed by an independent NATA accredited laboratory (Envirolab Services, Sydney, NSW, Australia) to confirm actual PFAS concentrations (Table 1).

### PFAS exposure regimen

A total of 216 four to five week-old male Swiss CD1 mice (*SwissTacAusb*, Swiss Webster Outbred), derived from the Taconic Swiss Webster lineage, were obtained from Australian BioResources (ABR; Moss Vale, NSW, Australia), or the University of Newcastle Animal Services Unit (NSW, Australia). We have complied with all relevant ethical regulations for animal use. As such, these mice were monitored, handled, and euthanized in accordance with the NSW Animal Research Act 1998, NSW Animal Research Regulation 2010, and the Australian Code for the Care and Use of Animals for Scientific Purposes 8th Edition, and with the ethical approval of the University of Newcastle Animal Care and Ethics Committee (A-2020-009). Upon receipt, mice were transferred to an *ad libitum* soy-free diet (Specialty Feeds, WA, Australia; SF06-053) to avoid exposure to estrogenic isoflavones, which might otherwise influence reproductive hormone profiles[68]. At four weeks of age, these mice were randomly assigned to one of three exposure groups (Fig. 1; 72 per group): control (untreated water), low, or high dose PFAS. Randomization was performed at the point of animal arrival by allocating individuals sequentially into control and treatment groups in an alternating manner to ensure balanced group sizes. This approach ensured an unbiased distribution of animals across groups while maintaining practical feasibility during the allocation process. Sample sizes were determined using G*Power, informed by pilot experiments and in consultation with statisticians. All researchers within the study were aware of the group allotment at all stages of the experiment, including allocation, exposure and data collection. Throughout the exposure period, mice were housed in cages of two to five per cage with 1.25 mm grit size corn-cob bedding (Corncobology Pty Ltd, Mt Kuring gai, Australia) bedding, a red-tinted hide, and wooden sticks provided as enrichment. To minimize potential confounders, experimental procedures were standardized across all groups. The order of treatments and measurements was alternated or counterbalanced where possible to avoid systematic bias. Animal handling was performed by the same researchers throughout, and cage positions were rotated regularly to control for potential location effects within the facility. PFAS water was prepared by first weighing each Hydropac film bags (Allied Scientific, Quebec, Canada) containing filtered water and then spiking it with an appropriate amount of stock PFAS working solution (i.e., 60 µL of a 1000× stock PFAS solution was injected into a 600 mL bag of water to prepare the low dose water or alternatively, 600 µL of the stock solution was injected into a 600 mL bag of water for the high dose exposure). Estimated PFAS consumption rates (mg/kg/day; described below) are provided in Table 1.

Following the twelve-week exposure period, mice were euthanized via $CO_2$ asphyxiation and weighed. Blood was immediately collected via cardiac puncture and centrifuged at $21,000 \times g$ for 10 min at 4 °C to separate the plasma for assessment of both circulating PFAS and hormone concentrations. Reproductive tissues were then harvested and either immediately assayed, fixed, or frozen, depending on the experiment to be performed. Tissue fixation was achieved by incubation in Bouin's fixative (9.0% formaldehyde, 5.0% acetic acid, 0.9% picric acid) for 24 h at 4 °C with constant rotation. Once fixed, tissues were washed twice in 50% ethanol for 24 h, and then subjected to another two 24 h washes in 70% ethanol before storage at 4 °C in fresh 70% ethanol until sectioning. Alternatively, tissues were frozen and kept at −80 °C until required.

### Analysis of blood plasma hormone profiles

Analysis of steroid hormone concentrations was performed on a subset of mice at the ANZAC Research Institute (Sydney, NSW, Australia) on 200 µL aliquots of thawed blood plasma using liquid chromatography and tandem mass spectrometry (LC-MS-MS) as previously described[69]. The limit of detection for each hormone is shown in Fig. 4. Hormones present above the limit of detection in more than two samples for each exposure group were subjected to either a one-way ANOVA with Tukey's multiple comparisons test (if normally distributed) or Kruskal–Wallis test (if not normally distributed).

## Quantitative analysis of PFAS in blood plasma and testes

After dissection, a fresh whole testis was weighed and then frozen as indicated above. Just prior to PFAS assessment, the testis was retrieved from storage, thawed and subsequently freeze-dried, weighted and homogenized. Homogenized samples were re-weighed and then transported on ice to Envirolab Services for quantitative analysis of PFAS using proprietary protocols. Envirolab Services were also employed to quantitatively assess PFAS in 200 μL samples of blood plasma. Briefly, extracted internal standards (EIS; 23 mass-labeled PFAS) were added during sample preparation to correct results for extraction and/or analytical effects that arise during the test workflow. For the purpose of quality control, all experimental samples were analyzed in tandem with a standard set of laboratory control samples. Prior to analysis, each matrix (i.e., water, serum, or testis) was prepared as appropriate. For water, each sample was methanol-amended and then filtered prior to the addition of the EIS. For plasma, after addition of EIS, samples were acid-crashed to remove complex biomolecules (e.g., proteins and phospholipids) that could otherwise interfere with analysis. Samples were then centrifuged, and the supernatant was subjected to weak anion exchange solid-phase extraction matrix separation. For the testis, sample extraction was achieved using basified methanol containing the EIS. Samples were then sequentially sonicated and agitated prior to EnviCarb cleanup, a process that involves the use of fine carbon to remove non-PFAS co-extracted organics. In all cases, the resulting extracts were analyzed by liquid chromatography-tanden mass spectrometry (LC-MS-MS) and the concentrations of the nine PFAS of interest are reported in Table 1. For the testis, results were normalized to the initial dry weight of each sample.

## Quantitative analysis of cholesterol in blood plasma and testes

Analysis of total cholesterol in mouse testicular tissue (10 mg testis tissue / 100 μl assay buffer II) and blood plasma (diluted 20-fold in assay buffer II) was conducted using a commercially available colorimetric reagent kit (Cholesterol Assay Kit; Abcam, UK; AB65359) as per the manufacturer's instructions.

## Bioaccumulation factor (BAF) and estimated daily intake values

BAF values, which quantify bioaccumulation of a substance within an organism from its environment, were calculated using the following equations and expressed as unitless.

$$\text{Plasma BAF} = [\text{PFAS in plasma}]/[\text{PFAS administered}] \quad (1)$$

$$\text{Testis BAF} = [\text{PFAS in testis}]/[\text{PFAS administered}] \quad (2)$$

In conducting these calculations, estimates of mouse daily PFAS intake were determined, taking into consideration the substrate consumption value based on the average daily water intake of an adult male mouse (~6 mL/mouse/day)[70]. These data were then imputed into the following equation, together with the average weight (49 g) of mice at euthanasia (i.e., maximal weight during experiment) and presented as mg/kg/day.

$$\text{Estimated daily PFAS intake} = \text{Daily substance consumption (mg)}/\text{weight (kg)} \quad (3)$$

As an extension of this analysis, estimates of equivalent PFAS intake for an adult human male exposed to an equivalent cocktail were determined by applying an average fluid intake of 3.7 L per day and an average weight of 87 kg[71,72]. These values were also used to calculate the safe drinking water values in mg/kg/day based on the reported values from the NHMRC[30], EPA Maximum Contaminant Levels[31] and the EU Drinking Water Directive[32] (see Table 1).

## Assessment of daily sperm production (DSP)

A portion of frozen testis from seven mice per group was removed and weighed in preparation for analysis of DSP using published protocols[33,34].

Briefly, after thawing the testis samples at room temperature, they were immersed in 5.0 mL of DSP solution (0.15 M NaCl, 0.1 mM sodium azide, and 0.05% (v/v) Triton X-100) and then sonicated with short pulses for a total of 80 s (16 cycles of 5 s on and 10 s off) in a beaker of ice. A 10 μL aliquot of the resultant cell suspension was then settled onto a hemocytometer, and the number of elongated spermatid heads counted. Using the initial testis weight and elongated spermatid count, total DSP was calculated per testis and subsequently divided by 4.84 to account for the number of days mouse spermatids spend in stages 14–16 of spermatogenesis[33,34].

## RT-qPCR analysis of steroidogenic enzyme transcripts

RT-qPCR was performed to assess the expression levels of relevant steroidogenic enzyme transcripts in testis tissue. For this purpose, total RNA was extracted from frozen testes of six mice per group using a RNeasy mini kit (cat # 74104; Qiagen; Hilden, Germany) in accordance with the manufacturer's protocol. The concentration of total RNA was assessed with a Nanodrop Lite Spectrophotometer (Thermo Fisher Scientific; Waltham, MA, USA) and RNA quality confirmed via resolution on a 1.0% (w/v) agarose gel stained with 0.001% SybrSafe (cat # S33102; Thermo Fisher Scientific).

Reverse transcription was achieved using a SuperScript VILO cDNA Synthesis Kit (cat # 11-754-050; Thermo Fisher Scientific) as per the manufacturer's instructions. Quantification of luteinizing hormone/choriogonadotropin receptor (*Lhcgr*), steroidogenic acute regulatory protein (*StAR*), cytochrome P450 11A1 (*Cyp11a1*), cytochrome P450 17A1 (*Cyp17a1*), 3beta-hydroxysteroid dehydrogenase/delta(5)-delta(4)isomerase type I (*Hsd3b1*), hydroxysteroid 17-beta dehydrogenase 3 (*Hsd17b3*), steroid 5-alpha-reductase 2 (*Srd5a2*), hydroxysteroid 17-beta dehydrogenase 6 (*Hsd17b6*) transcript abundance was undertaken using a LightCycler 96 (Roche; Basel, Switzerland) with a master mix comprised of 1× GoTaq master mix (cat # A6001; Promega; Madison, WI, USA), 0.15 μM each of forward and reverse primer and 5.0 ng/μL cDNA. Beta-actin (*beta-actin*) was used as the reference gene for relative quantification of gene expression using the delta c(t) method, with forward and reverse primers at a concentration of 0.25 μM (sequences in Supplementary Table 1).

## Assessment of apoptosis in germ cells via TUNEL (ApopTag)

Bouins fixed testes were sectioned at 4 μm and positioned onto slides. Slides were then dewaxed by three sequential incubations in xylene before rehydration in decreasing concentrations of ethanol (100, 90 and 70%). Antigen retrieval was achieved by incubation in proteinase K (20 μg/mL) for 15 min at room temperature, before two sequential washes in 1× phosphate buffered saline (PBS). An ApopTag Fluorescein in Situ Apoptosis Detection Kit (cat # S7110; Merck) was used to identify apoptotic cells via an indirect TUNEL method following the manufacturer's protocol. Following ApopTag application, slides were washed three times in PBS and then counterstained with 4′,6-diamidino-2-phenylindole (0.5 μg/mL) for 1 min at room temperature before being mounted in Mowiol containing 1,4-diazabicyclo[2.2.2]octane under a coverslip. All slides were imaged using an AXIOplan Imager 2 fluorescence microscope (Carl Zeiss Micro Imaging GmbH, Jena, Thuringia, Germany). Three biological replicates were analyzed for each exposure group, with the average number of TUNEL-positive cells recorded across 50 tubules per sample. An additional slide was treated with neat DNase I buffer (cat #M6101; Promega Madison, WI, USA) for 10 min, followed by incubation in DNase I enzyme and its buffer at a ratio of 1:1 for a further 10 min, as a positive control. Additionally, a negative control slide was prepared in which the TdT enzyme was omitted and instead substituted with PBS alone.

## Preparation of mouse spermatozoa

Mature mouse spermatozoa were isolated from the caudal segment of the epididymis, immediately after dissection, by retrograde perfusion via the vas deferens, as previously described[73]. Following isolation, spermatozoa were resuspended in 1.0 mL of pre-warmed BWW medium, at a concentration of 6–10 million cells/mL, in a flat bottom screw cap tube (Sarstedt, Numbrecht,

Germany). Depending upon the designated assay, cells were either assessed immediately or pelleted by centrifugation ($500 \times g$ for 3 min) to be fixed (4% paraformaldehyde for 15 min at room temperature) or snap-frozen and stored at $-80\,°C$ until required.

## Assessment of motility, viability, and reactive oxygen species status in sperm

Sperm motility parameters were assessed using CASA (IVOS; Hamilton Thorne; Beverly, MA, USA)[29]. Cells were loaded into pre-warmed 100 μm, two-chamber slides (Leja Products BV, Spoorzicht, Netherlands) and a minimum of 100 cells were assessed over five fields of view. The suite of motility parameters assessed included total motility, progressive motility, straight line velocity, curvilinear velocity, time-average velocity, linearity, straightness, amplitude of lateral head displacement, beat cross frequency, balancing, distance average path, distance curved line, and distance straight line. Sperm viability was assessed using a Sytox Green (SyG) flow cytometry-based assay (10 nM SyG at 37 °C for 15 min; cat # S7020; Invitrogen; Waltham, MA, USA)[29]. Using a FACS-Canto flow cytometer (BD Biosciences; San Jose, CA, USA), a minimum of 10,000 cells were assessed per sample and the resultant data analyzed using FACSDiva software (version 9.01; BD Biosciences). To determine if PFAS exposure elicited the generation of mitochondria-derived ROS in spermatozoa of PFAS-exposed mice, a MSR assay (Life Technologies cat # M36008) was used[29]. For this purpose, $1 \times 10^6$ fresh spermatozoa were resuspended in BWW and stained using 2.0 μM of MSR for 15 min at 37 °C. Cells were then centrifuged ($500\, x\, g$ for 3 min) and resuspended in 350 μL BWW for flow cytometric analysis.

## In vitro capacitation of mouse spermatozoa

The functional competence of spermatozoa isolated from PFAS-exposed mice was assessed by examining their ability to capacitate in vitro. Here, each sample was divided into two tubes and incubated in either non-capacitating BWW (NC-BWW prepared with 5.6 mM NaCl in place of $NaHCO_3$ to maintain an equivalent osmolarity) medium to suppress capacitation[29] or a modified complete BWW supplemented with 3.0 mM pentoxifylline and 5.0 mM dibutyryl cyclic adenosine monophosphate (cAMP); pharmacological agents that accelerate capacitation in vitro (CAP-BWW)[74]. Both groups were incubated for 45 min at 37 °C in 5.0% $CO_2$. Spermatozoa were then either fixed (as above) for immunolabeling with anti-phosphotyrosine antibodies (PT66; cat # P5872; Merck) or prepared for protein extraction and immunoblotting to determine their capacitation status using standard protocols[29]. Briefly, equivalent concentrations of extracted proteins were subjected to polyacrylamide gel electrophoresis (SDS-PAGE) using 4–20% Mini-PROTEAN gels (Bio-Rad, Cat# 4568095), before being transferred to a nitrocellulose membrane. Phosphotyrosine status, was examined by incubating blocked membranes (1 h at room temperature with 3.0% (w/v) BSA in Tris-buffered saline (20 mM Tris, 150 mM NaCl, pH 7.6) supplemented with 0.1% Tween (TBST)), with HRP-conjugated anti-phosphotyrosine antibodies diluted 1:4000 in 1.0% (w/v) BSA/TBST and for 1 h. Membranes were probed with enhanced chemiluminescence reagents (ECL plus, Amersham Bioscience) in accordance with the manufacturer's recommendations, and were visualized using a Bio-Rad ChemiDoc MP imaging system according to the manufacturer's protocol (Bio-Rad). To confirm equal protein loading, membranes were subsequently stripped for 40 min using Western Re-Probe reagent (G Biosciences, MO, USA; Cat# 786-306) and reprobed with anti-glyceraldehyde-3-phosphate dehydrogenase (GAPDH) antibodies (Merck; Cat# G9545) diluted 1:2000 in 1.0% (w/v) BSA/TBST for 1 h.

## Assessment of sperm DNA integrity

To determine the susceptibility of sperm DNA to PFAS induced damage, these cells were subjected to SCSA in which the acridine orange (AO; 300 μL of 1.0 mg/mL AO stock, 50 mL staining buffer (0.1 M citric acid, 0.2 M $Na_2HPO_4$, 1.0 mM EDTA, 0.15 M NaCl)) label differentiates sperm with densely compacted chromatin surrounding double stranded DNA (green fluorescence) from that of sperm with poorly compacted chromatin

surrounding single stranded DNA (red fluorescence)[29,75]. To this end, frozen spermatozoa samples were thawed on ice before being resuspended in 100 μL BWW medium at an approximate concentration of 10,000 cells/μL. This suspension was subsequently transferred to a flow cytometer tube containing 200 μL of acid–detergent solution (0.08 M HCl, 0.15 M NaCl, 0.1% Triton X-100) for 30 s followed by addition of 600 μL of AO staining solution before analysis. Analysis was carried out on a pre-equilibrated flow cytometer (10 min with a buffer comprising 25% acid–detergent solution and 75% AO staining solution). Additionally, an alkaline comet assay was used to identify the presence of DNA strand breaks within the spermatozoa of PFAS exposed male mice[29]. In this instance, frozen sperm pellets were thawed on ice and resuspended at a concentration of 4000 cells/μL in PBS. This sperm suspension was diluted 1:7 with pre-warmed CometAssay LMAgarose (R&D Systems; Cat# 4250-050-02) and then spread onto frosted slides (Trajan, Victoria, Australia) pre-coated with 1.0% low melting point agarose (Cat# A4018; Merck), and a coverslip applied. Once the agarose had solidified (4 °C for a minimum of 1 h), the slides were treated with lysis solution 1 (0.8 M Tris, 0.8 M dithiothreitol (DTT), 1.0% SDS, pH 7.5) for 30 min at room temperature followed by lysis solution 2 (0.4 M Tris, 50 mM EDTA, 2.0 M NaCl, 0.4 M DTT, pH 7.5) for 30 min at room temperature. Slides were washed in Tris–boric acid–EDTA (TBE) solution (0.4 M Tris–HCl, 0.4 M boric acid, 10 mM EDTA, pH 7.5) for 10 min at room temperature and then incubated for 15 min with alkaline solution (0.03 M NaOH, 1.0 M NaCl, pH 11.5) at 4 °C in preparation for electrophoresis (1 V/cm for 4 min). Following electrophoresis, slides were washed for 5 min in neutralization solution (0.4 M Tris, pH 7.5) at room temperature. Immediately prior to viewing, each slide was incubated in SYBR Green nucleic acid stain (Lonza, Rockland, ME, USA; Cat# 50513) diluted 1:10,000 with Tris EDTA (10 mM Tris, 1.0 mM EDTA, pH 8). DNA fragmentation was determined using Comet Assay IV software (Perceptive Instruments, Suffolk, UK), in which the relative fluorescence intensity of the comet tail was used as a measure of the degree of DNA damage, normalized to the untreated control. A minimum of 50 cells were analyzed per slide.

## Zona pellucida binding, in vitro fertilization (IVF) and embryo development

To assess the zona pellucida binding ability of PFAS-exposed spermatozoa, unexposed Swiss CD1 female mice aged 4–6 weeks were super-ovulated following an intraperitoneal injection regime comprising 7.5 IU equine chorionic gonadotropin and 7.5 IU human chorionic gonadotropin (hCG) (Intervet, Sydney NSW, Australia)[76,77]. MII oocytes were harvested from the ampulla 13–15 h following hCG and incubated in 300 μg/ml hyaluronidase at 37 °C for 1–3 mins to remove cumulus cells. Any remaining cumulus cells were removed by a further 3–5 washes in M2 medium. Isolated oocytes were subsequently stored in a high salt storage media (1.5 M $MgCl_2$, 0.1% (w/v) dextran, 10 mM HEPES buffer, 0.1% (w/v) PVA at pH 7.3) at 4 °C until use. Prior to incubation with spermatozoa, the oocytes were washed five times with appropriate BWW media (either NC-BWW or CAP-BWW, depending on the sperm sample to be added).

Spermatozoa were collected fresh and either driven to capacitate as previously described or resuspended in NC-BWW media (1 million cells/mL) as a negative control. For all samples, 20,000 sperm cells were added to a droplet containing 8–10 oocytes and left for 15 min under mineral oil at 37 °C. To ensure experimental validity, a motility count was performed during gamete co-incubation. At the completion of the co-incubation period, oocytes were gently washed by pipetting up and down into fresh medium to remove any unbound or loosely adherent spermatozoa. A coverslip with Vaseline/paraffin posts was then gently placed onto the slide and lowered into position, enabling the number of sperm bound to each zona pellucida to be recorded using phase contrast microscopy.

To assess the competency of spermatozoa from PFAS-exposed males to initiate and support early embryonic development, mature cumulus oocyte complexes were collected from unexposed females following superovulation, as described above[78]. Mature MII oocytes were identified by the presence of a single polar body and the absence of a germinal vesicle.

Spermatozoa isolated from each cohort of mice were subsequently incubated in BWW supplemented with 1.0 mg/mL polyvinyl alcohol and 1.0 mg/mL methyl-β-cyclodextrin for 45 min at 37 °C to permit capacitation[77,79,80]. Thereafter, 20,000 spermatozoa were co-incubated with oocytes for 4 h at 37 °C in 5.0% $O_2$, 6.0% $CO_2$ in $N_2$. Extrusion of the second polar body and/or pronucleus formation were then examined as signs of successful fertilization. Embryos were cultured for 96 h[81] and the percentage of fertilized oocytes and reached the blastocyst stage was calculated.

## Sperm RNA extraction and small-RNA-sequencing
Total RNA was extracted from populations of mature spermatozoa from which sncRNAs (18-40 nucleotides) were then purified as previously described[82]. Purified sncRNAs were used to generate sncRNA libraires via ligation-dependent sncRNA library construction, using a modified Illumina TruSeq Small RNA Library Prep Kit (cat #20005613, Illumina; San Diego, CA, USA). Cloning was performed as per the manufacturer's instructions. Small RNA libraries were quantified and combined at equimolar concentrations and sequenced on a NextSeq1000 instrument using 50 base pair (bp) paired-end chemistry and analyzed as previously described[62,82].

## Embryo mRNA-sequencing
Groups of 4-cell embryos ($n$ = 3–4 replicates with 8–10 embryos/replicate) fertilized by the spermatozoa of control or PFAS-exposed males were transferred to microcentrifuge tubes in 1 × TCL buffer (cat # 1070498; Qiagen, Hilden, Germany) supplemented with 1.0% β-mercaptoethanol and stored at −80 °C until sequencing. Embryo RNA-seq was conducted using a SMART-Seq protocol as previously described[35]. Briefly, RNA was isolated from embryo samples using RNAClean-XP-SPRI beads (Beckman Coulter, Cat # A63987) and cDNA generated using Superscript II (cat # 18080044; Invitrogen) and amplified for 10 cycles. Tn5-mediated tagmentation was used with a Nextera XT kit (cat # FC-131-1096; Illumina) to add adapters for indexing and sequencing. Pooled libraries were sequenced on a NextSeq1000 instrument in house. Bioinformatic pipelines were assembled using DolphinNext[83]. Data were mapped to the *Mus musculus* genome (mm10) using RSEM[82]. One control embryo group was excluded from analysis as it had fewer than 9000 detected transcripts. Raw count data were imported into R Statistical Software and counts were normalized and analyzed for DEGs using DESeq2 package[84]. Genes were considered DEGs when the $p < 0.05$ and fold-change ±1.5.

## Bioinformatic assessment of RNA-sequencing data
IPA software (Qiagen) was used to analyze the refined list of DEGs from 4-cell embryos as previously described[62]. Canonical pathways, upstream regulators, and disease(s) they putatively influence were assessed using the returned $p$ values and $Z$-scores. The predicted extent of dysregulation of pathways, regulators and diseases is denoted by a $p$ value of ≤ 0.05 and a Z-score, whereby any output above a threshold of ≥2 is considered "activated", while conversely, any output ≤−2 is predicted to be "inhibited".

## Statistics and reproducibility
All statistical analyses and graph generation were performed using GraphPad Prism 9 software (Dotmatics; Boston, MA, USA). Data were first tested for normality before being subjected to either a one-way or two-way ANOVA with Tukey's multiple comparisons test or the Kruskal-Wallis test with Dunn's multiple comparisons test, as appropriate. All IVF data involving proportions were arcsine transformed prior to statistical analyses. Data are presented as the mean ± SEM calculated based on a minimum of three biological replicates per exposure group per assay, with a $p < 0.05$ considered statistically significant. Biological replicates are defined as individual male mice.

## Reporting summary
Further information on research design is available in the Nature Portfolio Reporting Summary linked to this article.

## Data availability
The gene expression datasets generated and analyzed during the current study are available in the Gene Expression Omnibus repository with accession number GSE271479. Further data that support the findings of this study, including the numerical source data for graphs and charts are available in the Supplementary Data 5. Any additional information is available from the corresponding author upon reasonable request.

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

## Acknowledgements

We would like to thank our colleagues at the Hunter Medical Research Institute for the preparation of histological slides and the Animal Services Unit at the University of Newcastle. We also wish to thank our grant support. This work was supported by funding from the National Health and Medical Research Council of Australia (NHMRC) Targeted Call for Research into Per- and Poly-Fluoroalkylated Substances (APP1189415) awarded to B.N., M.P.G., G.N.D., M.D.D., B.T., A.L.E., and S.D.R. B.N. is also the recipient of an NHMRC Senior Research Fellowship (APP1154837) and M.D.D. is the recipient of an NHMRC Investigator Grant (APP1173892) and a Defeat DIPG ChadTough New Investigator Fellowship. This work was also supported by an Australian Government Research Training Program (RTP) Scholarship awarded to L.G.

## Author contributions

B.N., M.P.G., G.D.I., M.D.D., S.D.R., B.D.T., and A.L.E. conceived the study, sourced funding, contributed to experimental design and data interpretation. Experimentation and data analysis was performed by L.G., J.H.M., A.L.A., I.R.B., S.J.S., N.A.T., J.E.S., S.P., S.P.S., C.C.C., R.D., A.G., and D.J.H. L.G., J.H.M. and B.N. drafted the manuscript. All authors edited the manuscript and approved the final version.

## Competing interests

The authors declare no competing interests.

## Additional information

[1]School of Environmental and Life Sciences, University of Newcastle, Callaghan, NSW, Australia. [2]Infertility and Reproduction Research Program, Hunter Medical Research Institute, New Lambton Heights, Newcastle, NSW, Australia. [3]Division of Neonatology, Children's Hospital of Philadelphia, Philadelphia, PA, USA. [4]School of BioSciences, Faculty of Science, University of Melbourne, Melbourne, VIC, Australia. [5]Departments of Genetics and Pediatrics - Penn Epigenetics Institute, Institute of Regenerative Medicine, and Center for Reproduction and Women's Health, Perelman School of Medicine, University of Pennsylvania, Philadelphia, PA, USA. [6]ANZAC Research Institute, University of Sydney, Sydney, NSW, Australia. [7]School of Health, University of the Sunshine Coast, Maroochydore, QLD, Australia. [8]Cancer Signalling Research Group, School of Biomedical Sciences and Pharmacy, College of Health, Medicine and Wellbeing, University of Newcastle, Callaghan, NSW, Australia. [9]Precision Medicine Research Program, Hunter Medical Research Institute, New Lambton Heights, Newcastle, NSW, Australia. [10]Centre for Technology in Water and Wastewater, School of Civil and Environmental Engineering, University of Technology Sydney, Ultimo, NSW, Australia. [11]Priority Research Centre for Geotechnical Science and Engineering, University of Newcastle, Callaghan, NSW, Australia. [12]NSW Health Pathology, Newcastle, NSW, Australia. [13]These authors contributed equally: Leah Gillespie, Jacinta H. Martin. ✉e-mail: jacinta.martin@newcastle.edu.au

