## [Transparent Peer Review file · Communications Biology]

Exposure of mice to environmentally relevant per- and polyfluoroalkyl substances (PFAS) alters the sperm epigenome

Corresponding Author: Dr Jacinta Martin

Version 0:

Reviewer comments:

Reviewer #1

(Remarks to the Author)

The authors investigated the effects of low and high per- and polyfluoroalkyl substances (PFAS) treatment on male reproductive function in mice. After twelve weeks of continuous exposure, they assessed the physiological function of the testis, sperm, and hormone levels in blood, as well as the small non-coding RNA profiles of sperm and gene expression during early embryonic development. While the study is relatively comprehensive in phenotypic assessment, there are several concerns regarding the interpretation of the results.

Major concerns:

1. The study reports that daily sperm production is reduced in the low PFAS group, whereas it remains similar in the high PFAS group compared to the control group. This observation needs further explanation to clarify this phenomenon. Also, it appears that protein phosphorylation levels are higher in the low PFAS group compared to both the control and high PFAS groups, with GAPDH used as a loading control. The authors should clarify whether GAPDH is a suitable loading control in this context and whether its expression remains consistent under PFAS treatment.
2. The small RNA analysis was performed using traditional small RNA sequencing methods, which are known to be biased and incapable of detecting certain small RNAs, including highly modified small RNAs, such as tRNA-derived small RNAs (tsRNAs) and rRNA-derived small RNAs (rsRNAs). The authors are encouraged to consider utilizing a more suitable approach, such as PANDORA-seq (PMID: 33820973), which would allow for a more comprehensive analysis of the small RNA landscape and potentially reveal additional changes between the control and PFAS groups.
3. The manuscript does not address rsRNAs and mRNA-derived small RNAs, despite notable changes that seem to occur with increasing PFAS levels. The authors should consider including these analyses to provide a more complete overview of how PFAS exposure affects small RNA profiles.
4. The manuscript lacks references to recent studies on environmental toxicants that disrupt small RNA profiles and induce phenotypic transmission to offspring (e.g., PMID: 37691128, 36709676). Including these references would provide better context for the study and demonstrate an awareness of current research in this area.

Minor concerns:

Table 1 is currently divided across six pages. It would be more reader-friendly if the table were consolidated into a single page or provided as an Excel file to improve accessibility and readability.

Reviewer #2

(Remarks to the Author)

In this manuscript, Calvert et al. investigated the impact of PFAS on male sex hormone levels and sperm epigenome in mice. The mice were exposed to low or high doses of PFAs through drinking water. They found that with both low and high dosage of PFAs, there is an accumulation of PFAs detected in the plasma and testes, though testes had a lower

concentration. They also found significant changes in the low dosage group in the main male androgens. In the high dosage group, they only observed changes in corticosterone. They further investigated the testis samples and analyzed the sperm of the mice and discovered that mice administered with a low dose of PFAs had decreased daily sperm production, despite the lack of increased apoptosis. They also found that PFAS exposure did not affect sperm parameters including viability, motility, and ability to penetrate the zona pellucida of oocytes. The authors then analyzed the small non-coding RNA profile of the sperm and found that high PFA-exposed males had significantly altered sperm miRNAs. They then investigated if sperm sncRNA can affect embryonic gene expression. Embryos were fertilized with PFAS-exposed sperm and many differentially expressed genes were altered. The manuscript is well-written and includes comprehensive experiments. I have the following comments for editing:

1. One of the findings was a reduction in seminal vesicle fluid: body weight ratio in PFA treated mice. The authors may want to elaborate on this finding and discuss the potential significance. .
2. Did PFAS exposure affect the cholesterol levels in the mice? Studies have indicated the link of testosterone levels to hypercholesterolemia and cholesterol is also the precursor of many hormones including testosterone. It would be interesting to know if PFAS exposure affects circulating or reproductive organ cholesterol contents.
3. Figure 4 is visually cluttered and is a bit difficult to follow. The authors should consider reformatting this figure or split the figure to two figures.
4. The authors should also consider discussing the limitations of current small RNA sequencing methods as more and more studies have demonstrated that miRNAs are not major type of sncRNAs in sperm and other types of sncRNA such as tRFs play a more important role in mediating embryonic gene expression and epigenetic inheritance.
5. There is a switch from passive to active writing voice in the section Methods---Bioaccumulation factor (BAF) and estimated daily intake values. It would be better to maintain consistency in passive voice in the methods section.

Reviewer #3

(Remarks to the Author)

1) General comments

This manuscript presents a study on the impact of PFAS exposure on male reproductive function in mice. The findings contribute to the understanding of environmental influences on reproductive health. The paper is generally well-written and informative; however, there are several areas that require significant revisions to enhance clarity and depth.

2) Specific comments

a) Major concerns

1. While the study reports that PFAS exposure reduced sperm production and altered the sperm's sncRNA profile, it simultaneously claims that the exposed sperm are "morphologically and functionally normal." However, these contradictory conclusions are not adequately explained. For instance, changes in sncRNA should theoretically have an impact on sperm function, but no clear link between these molecular changes and functional outcomes is drawn. Reduced sperm production and molecular alterations often correlate with impaired sperm quality, yet the study fails to provide a cohesive biological explanation for these discrepancies.
2. The discussion highlights alterations in sperm sncRNA and suggests that these may cause abnormal gene expression in early embryos, but it does not provide direct evidence for how these changes translate into specific physiological or pathological outcomes. For instance, it only speculates that these changes may relate to "body size" and "development of the hematopoietic system," without offering concrete phenotypic assessments to support the claim that these molecular changes will lead to functional or clinical consequences.
3. The study suggests that PFAS exposure may impair the hypothalamic-pituitary-testicular (HPT) axis and interfere with steroid hormone synthesis by inhibiting key enzymes. However, the authors only measure the expression levels of a few related genes, without conducting a detailed functional analysis of the HPT axis or measuring the activity of key enzymes. This makes the mechanistic hypothesis overly simplistic, lacking the robust experimental data necessary to substantiate these claims.

b) Minor concerns

1. Although the study observed that low-dose PFAS exposure had a more pronounced effect on hormone levels than high-dose exposure, the explanation for this phenomenon remains superficial, relying on the literature's reference to "nonlinear dose-response" patterns. There is no in-depth discussion of why this dose-response relationship is so marked in this study, leaving the interpretation of these results incomplete and somewhat superficial.
2. The study hypothesizes that PFAS exposure may affect offspring health and development via alterations in sperm sncRNA, but no experiments were conducted to directly test these transgenerational effects. This weakens the argument, as the conclusion about potential transgenerational impacts is speculative and needs further experimental evidence to be convincingly established.
3. Inconsistent Figure Sizing and Table Formatting: Figure sizes, such as in Figure 8, lack consistency, which affects the overall presentation quality of the manuscript. Additionally, several tables contain incomplete information, which hinders clarity. The authors should ensure consistent figure sizing and optimize the layout of the tables to present a more polished and comprehensive manuscript.

Version 1:

Reviewer comments:

Reviewer #1

(Remarks to the Author)

1. The authors defend GAPDH by normalizing it to tubulin/HSPA2 and conclude all are stable. But if PFAS (or capacitation state) alters multiple housekeeping proteins, cross-normalization can mask concerted shifts; what they need is a total-protein load control or an external spike-in.
2. Please also use nomenclature in the manuscript text and figures—tRNA-derived small RNAs (tsRNAs), rRNA-derived small RNAs (rsRNAs), and mRNA-derived small RNAs (msRNAs)—in parallel with tRFs and rRFs. Limiting these terms to the response letter for reviewer only reduces discoverability.

Reviewer #2

(Remarks to the Author)

The authors have adequately addressed previous comments. The revised manuscript is now suitable for publication.

Reviewer #3

(Remarks to the Author)

This paper provides a comprehensive assessment of the impacts of environmentally relevant PFAS mixtures on male reproductive function in mice. The authors present valuable findings on reduced sperm production and altered sperm epigenetics linked to early embryonic gene dysregulation. While some results align with prior studies, the integration of sperm function, epigenetic changes, and embryo development offers novel insights. The conclusions are adequately supported by experimental data, though additional research could further strengthen causal links. The statistical methods are appropriate, and methodological details enable reproducibility, despite minor ambiguities. Overall, this well - written paper contributes significant data to the field of reproductive toxicology and environmental health, highlighting male reproductive vulnerability to PFAS. It has the potential to influence future research and policies, and thus, I recommend its acceptance.

Jun Wei Pek, PhD
Editorial Board Member
Communications Biology

MS#: COMMSBIO-24-5715-T

RE: Response to Reviewers

Dear Professor Pek,

On behalf of my co-authors, I wish to thank you and the Reviewers for your thorough review of our manuscript entitled "*Exposure of mice to environmentally relevant per- and polyfluoroalkyl substances (PFAS) alters the sperm epigenome*" (MS#: COMMSBIO-24-5715-T) and the opportunity you have provided for us to submit a revised version of the manuscript in which we address their concerns. We have carefully considered and responded to all of the Reviewer comments and believe that our manuscript has been substantially improved by addressing their insightful suggestions. Accordingly, below we have included a response letter in which we provide point-by-point answers to the reviewers' comments. In addition, we have submitted both a clean copy of the revised manuscript and a marked-up copy in which our changes have been highlighted in yellow. Please note that the page/line numbers listed below in our Response to Reviewers reflect those in the marked-up (as opposed to clean) version of the revised manuscript. We remain committed and enthusiastic to publish this study in Communications Biology and we hope that you will view our revised article with equal enthusiasm.

All best wishes,

Jacinta

RESPONSE TO REVIEWERS

EDITORIAL BOARD MEMBER

1. Your manuscript entitled "Exposure of mice to environmentally relevant per- and polyfluoroalkyl substances (PFAS) alters the sperm epigenome" has now been seen by 3 referees, whose comments are appended below. You will see from their comments copied below that while they find your work of potential interest, they have raised quite substantial concerns that must be addressed. In light of these comments, we cannot accept the manuscript for publication but would be interested in considering a revised version that addresses these serious concerns.

Answer: We wish to thank you for recognizing the merits of our study and providing the opportunity to address the Reviewer's comments in a revised version of our manuscript. Below, we provide point-by-point answers to each of the reviewer's comments and highlight (in yellow) the changes that have been made to our manuscript to address their concerns.

2. We think that PANDORA-seq is not essential for publication here, but it is encouraged to address the RNA-seq limitations as suggested by Reviewer 2.

Answer: Thank you for clarifying that we are not required to reperform small RNA sequencing using PANDORA-seq methodology as suggested by Reviewer 1. We have

taken your advice to instead acknowledge and discuss the limitations of the traditional RNA-seq strategy that was used in this study. Please see answers provided below in relation to Reviewer 1 Question 1 and Reviewer 2 Question 4. Please also refer to the inclusion of additional text in lines 883-887 of the revised manuscript.

3. Also, it is not essential to perform additional in-depth phenotypic analyses on the offspring for publication here, although it would be nice to have some.

Answer: Thank you for clarifying that we are not required to perform additional in-depth phenotypic analyses on the offspring. We acknowledge that this information would be beneficial, however we agree that such detailed analyses fall beyond the scope of the present study. In recognition of the importance of these analyses, we are currently designing a separate study to investigate the impact of PFAS exposure across multiple generations, which would enable us to conduct the suggested in-depth phenotypic analyses of offspring health across each generation.

4. You are advised to refer to the ARRIVE 2.0 guidelines for best practice on reporting *in vivo* methods (<https://arriveguidelines.org/arrive-guidelines>) and I encourage you to include a completed author checklist (<https://arriveguidelines.org/resources/author-checklists>) as a Supplementary file.

Answer: Thank for, we have taken your advice to refer to the ARRIVE 2.0 guidelines for best practice on reporting *in vivo* methods and have included a completed author checklist as a Supplementary file as requested.

5. If you decide to submit a revised version, we ask that you ensure your manuscript complies with our editorial policies. Please see our revision checklist for guidance on formatting the manuscript and complying with our policies. A comprehensive guide to our formatting requirements for final submissions is also available for your reference here.

Answer: We remain committed and enthusiastic about publishing this study in Communications Biology. Accordingly, below we have included a point-by-point response to all comments received from the three Reviewers. We have also adhered to the revision checklist guide in relation to manuscript formatting and compliance with journal policies.

6. Communications Biology is committed to improving transparency in authorship. As part of our efforts in this direction, we are now requesting that all authors identified as ‘corresponding author’ create and link their Open Researcher and Contributor Identifier (ORCID) with their account on the Manuscript Tracking System prior to acceptance. ORCID helps the scientific community achieve unambiguous attribution of all scholarly contributions. You can create and link your ORCID from the home page of the Manuscript Tracking System by clicking on ‘Modify my Springer Nature account’ and following the instructions in the link below. Please also inform all co-authors that they can add their ORCID to their accounts and that they must do so prior to acceptance. <https://www.springernature.com/gp/researchers/orcid/orcid-for-nature-research>

Answer: Thank you, the corresponding author confirms that they have linked their ORCID with their account on the Manuscript Tracking System.

REVIEWER #1

1. While the study is relatively comprehensive in phenotypic assessment, there are several concerns regarding the interpretation of the results.

Answer: Thank you for these comments, we trust that the changes we have made to the revised manuscript, and summarized below, are satisfactory to address all of this Reviewer's concerns.

2. The study reports that daily sperm production is reduced in the low PFAS group, whereas it remains similar in the high PFAS group compared to the control group. This observation needs further explanation to clarify this phenomenon. Also, it appears that protein phosphorylation levels are higher in the low PFAS group compared to both the control and high PFAS groups, with GAPDH used as a loading control. The authors should clarify whether GAPDH is a suitable loading control in this context and whether its expression remains consistent under PFAS treatment.

Answer: Thank you for these insightful comments. We agree that the compromise of daily sperm production achieved following the administration of a low- versus high-dose PFAS cocktail was an unexpected result and is not one for which we have any direct data to identify the biological mechanism. However, based on a thorough literature review of toxicological data, such findings are not without precedent. For instance, it has been reported that cognitive function in older adults (PMID: 33068581) and the prevalence of diabetes (PMID: 36964247 and PMID: 35970987) exhibits a non-monotonic dose-response with plasma PFAS. In the absence of experimental data to substantiate this mechanism, we had refrained from speculating on the phenomenon in our original article. However, in view of this comment and that raised by Reviewer 3 (see Reviewer 3 Question 5 below), we have now taken the opportunity to amend the Discussion to include additional context and a brief consideration of possible explanations. Please see lines 782- 789.

In relation to the suitability of GAPDH as an appropriate loading for immunoblotting analyses, we note that this house keeping protein is frequently used as a loading control in the context of sperm biology research (e.g. PMID: 27857155 and PMID: 34213690) and in PFAS research (e.g. PMID: 36326898, PMID: 38549690, PMID:38953992, PMID: 38117326). To the best of our knowledge there is no literature to suggest that the abundance of this protein may be specifically affected by PFAS exposure. However, this is certainly an important consideration and one that we have now sought to confirm under our own experimental conditions. Specifically, we have directly compared the relative abundance of GAPDH versus that of other commonly used protein loading controls such as alpha-tubulin (Tubulin) and the chaperone protein Heat Shock Protein Family A (Hsp70) member 2 (HSPA2) across a similar panel of samples. A summary of the densitometric data from this analysis is included in Figure 1 below for the Reviewer's consideration. Our interpretation of these data is that they confirm the suitability of GAPDH as a loading control since its abundance, relative to that of the other loading control proteins investigated, remains consistent under PFAS treatment.

Figure 1: Densitometric analysis of housekeeping proteins in sperm proteins lysates. Mouse spermatozoa were allocated into three treatment groups (control, low

PFAS exposure and high PFAS exposure) and incubated in either non-capacitating or capacitating BWW media as described in the accompanying manuscript. After incubation, equivalent amounts of sperm protein lysates (7.5 μ g) were resolved by SDS-PAGE and prepared for immunoblotting with either anti-GAPDH, anti-tubulin, or anti-HSPA2 antibodies, followed by appropriate HRP-conjugated secondary antibodies. Densitometric measurement of each target band was conducted and GAPDH densitometric data are presented normalised to either tubulin or HSPA2 (mean values \pm SEM, based on four biological replicates). No statistical difference in the relative amount of each house keeping protein was detected across all treatment groups, thus demonstrating the appropriateness of GAPDH as a loading control in this study.

3. The small RNA analysis was performed using traditional small RNA sequencing methods, which are known to be biased and incapable of detecting certain small RNAs, including highly modified small RNAs, such as tRNA-derived small RNAs (tsRNAs) and rRNA-derived small RNAs (rsRNAs). The authors are encouraged to consider utilizing a more suitable approach, such as PANDORA-seq (PMID: 33820973), which would allow for a more comprehensive analysis of the small RNA landscape and potentially reveal additional changes between the control and PFAS groups.

Answer: Thank you for highlighting these important considerations in relation to the limitations of the traditional small RNA sequencing methods used herein. We readily concede the limitations of such methods, but we regret that we are not in a position to recommence PFAS administration and collect the requisite material required to perform alternative PANDORA-seq analyses. Rather, we have taken the Editor's advice to modify our Discussion to acknowledge the limitations associated with the traditional RNA-seq method used herein, and the prospect that additional changes in the sperm small RNA landscape may have been detected had we used a more comprehensive approach such as PANDORA-seq. Please see lines 883-887.

4. The manuscript does not address rsRNAs and mRNA-derived small RNAs, despite notable changes that seem to occur with increasing PFAS levels. The authors should consider including these analyses to provide a more complete overview of how PFAS exposure affects small RNA profiles.

Answer: This is an excellent suggestion, and we offer our apology for not giving more consideration to these potentially important changes among the overall response of the sperm small RNA landscape to the imposed PFAS challenge. By way of explanation, we

originally elected to focus on the sncRNA subclasses for which there is better annotation regarding the functional implications of changing abundance (e.g. miRNAs, tRFs and piRNAs). However, in view of the Reviewer's comments, we have now reframed our results section to give additional consideration to PFAS-induced changes in rsRNAs (rRNA fragments) and mRNA-derived small RNAs to include a more complete overview of how PFAS exposure affects small RNA profiles. Please see lines 667-671 and Supplementary Fig. S4

5. The manuscript lacks references to recent studies on environmental toxicants that disrupt small RNA profiles and induce phenotypic transmission to offspring (e.g., PMID: 37691128, 36709676). Including these references would provide better context for the study and demonstrate an awareness of current research in this area.

Answer: We thank the reviewer for drawing our attention to these excellent studies, which reinforce the concept that environmental toxicants can disrupt sperm small RNA profiles and induce phenotypic transmission to offspring. We have amended our Discussion to include reference to the findings of the recommended studies. Please see lines 866 – 870.

6. Table 1 is currently divided across six pages. It would be more reader-friendly if the table were consolidated into a single page or provided as an Excel file to improve accessibility and readability.

Answer: We apologize for these formatting errors, which appear to have arisen during the automated conversion of our submitted files into a pdf. The original Table was submitted as an Excel spreadsheet, which should enable the Copy Editor to reformat the Table in accordance with the journal specifications.

REVIEWER #2

1. One of the findings was a reduction in seminal vesicle fluid: body weight ratio in PFA treated mice. The authors may want to elaborate on this finding and discuss the potential significance.

Answer: This is an excellent suggestion, however other than the recorded weights, we have no experimental data to substantiate the functional consequences of such changes in seminal vesicle fluid : body weight ratios. We are acutely aware of the importance of the seminal vesicles and their secretions in terms of promoting sperm function and influencing the immune environment within the female reproductive tract and we anticipate that the observed changes may therefore have consequences for *in vivo* fertilization and/or successful pregnancy outcomes. However, without substantiating experimental data we have deliberately refrained from speculating on the consequences of a change in weight ratio. We will prioritize such analyses in future work. We do hope the reviewer understands our position.

2. Did PFAS exposure affect the cholesterol levels in the mice? Studies have indicated the link of testosterone levels to hypercholesterolemia and cholesterol is also the precursor of many hormones including testosterone. It would be interesting to know if PFAS exposure affects circulating or reproductive organ cholesterol contents.

Answer: The reviewer raises an interesting question in relation to the influence of PFAS exposure on cholesterol abundance. Regrettably this analysis did not form part of our original study, however, in consideration of this excellent suggestion, we have now conducted an analysis of cholesterol levels within the testes and blood plasma of PFAS exposed mice compared to that of their control counterparts. The most pronounced finding of this analysis was a significant reduction in cholesterol levels within the testes of PFAS exposed animals, which was not evident in equivalent blood samples. Such findings could at least, in part, account for the accompanying reduction in male hormone levels, although we acknowledge that further studies are needed to determine the causal nature of this relationship. Please see revised Figure 4, panels O and P and lines 269-273 (methods) and 573-582 (results).

Calvert et al. Figure 4

3. Figure 4 is visually cluttered and is a bit difficult to follow. The authors should consider reformatting this figure or split the figure to two figures.

Answer: We thank the reviewer for this comment and acknowledge that the formatting of this figure to include all relevant data did present a number of challenges. Accordingly, we have taken their advice to split Figure 4 into two figures: (i) a new supplementary figure that outlines the hormone synthesis pathway and identifies the catalytic role of the relevant enzymes (please see Supplementary Figure Supplementary S1 and lines 558-561), and (ii) data quantify the circulating levels of a panel of steroid hormones and cholesterol (please see revised Figure 4).

4. The authors should also consider discussing the limitations of current small RNA sequencing methods as more and more studies have demonstrated that miRNAs are not major type of sncRNAs in sperm and other types of sncRNA such as tRFs play a more important role in mediating embryonic gene expression and epigenetic inheritance.

Answer: Here, the reviewer raises similar concerns to those identified by Reviewer 1 in relation to RNA sequencing methods (please see Reviewer 1, comment 3) and the potential importance of changes in the abundance of different subclasses of small RNA beyond that of miRNAs (please see Reviewer 1, comment 4). In response, we have included additional analysis of alternative small RNA subclasses (please see lines 667 – 671 and Supplementary Figure S4) and also amended the text of our Discussion to

acknowledge the limitations of the traditional RNA sequencing strategy used herein (please see lines 883 - 887).

5. There is a switch from passive to active writing voice in the section Methods--- Bioaccumulation factor (BAF) and estimated daily intake values. It would be better to maintain consistency in passive voice in the methods section.

Answer: Thank you for identifying this issue, which has now been resolved to maintain consistency in passive voice throughout the Methods section (please see lines 276 - 293).

REVIEWER #3

1. The paper is generally well-written and informative; however, there are several areas that require significant revisions to enhance clarity and depth.

Answer: Thank you for these comments, we trust that the changes we have made to the revised manuscript, and summarized below, are satisfactory to address all of this Reviewer's concerns.

2. While the study reports that PFAS exposure reduced sperm production and altered the sperm's sncRNA profile, it simultaneously claims that the exposed sperm are "morphologically and functionally normal." However, these contradictory conclusions are not adequately explained. For instance, changes in sncRNA should theoretically have an impact on sperm function, but no clear link between these molecular changes and functional outcomes is drawn. Reduced sperm production and molecular alterations often correlate with impaired sperm quality, yet the study fails to provide a cohesive biological explanation for these discrepancies.

Answer: Thank you for identifying the apparent inconsistencies in our conclusions. By way of explanation, we view changes to the sperm sncRNA profile and reduced daily rates of sperm production as 'cryptic phenotypes' since they have the potential to be overlooked during conventional clinical analysis of semen samples. Moreover, in our study these phenotypes did not appear to be correlated with conventional measures of impaired sperm quality (i.e. impaired motility, morphological deformities, or DNA damage). Nevertheless, in view of the Reviewer's excellent comments, we now appreciate that our explanations could readily be misinterpreted. Accordingly, we have carefully revised all text to remove any potentially ambiguous conclusions. Please see lines 758, 855, 859 – 860, 917, 925 - 929.

3. The discussion highlights alterations in sperm sncRNA and suggests that these may cause abnormal gene expression in early embryos, but it does not provide direct evidence for how these changes translate into specific physiological or pathological outcomes. For instance, it only speculates that these changes may relate to "body size" and "development of the hematopoietic system," without offering concrete phenotypic assessments to support the claim that these molecular changes will lead to functional or clinical consequences.

Answer: We readily acknowledge that embryonic / offspring phenotypic data would be beneficial in terms of validating our conclusions, however we agree with the Editor that that such detailed analyses regrettably fall beyond the scope of the present study. In recognition of the importance of these analyses, we are currently designing a separate

study to investigate the impact of PFAS exposure across multiple generations, which would enable us to conduct the suggested in-depth phenotypic analyses of offspring health across each generation. In acknowledgment of these limitations we have revised our Discussion to indicate that further studies and additional phenotypic assessments are necessary to validate the functional consequences of the observed molecular changes in sperm and embryos. Please see lines 925 – 929.

4. The study suggests that PFAS exposure may impair the hypothalamic-pituitary-testicular (HPT) axis and interfere with steroid hormone synthesis by inhibiting key enzymes. However, the authors only measure the expression levels of a few related genes, without conducting a detailed functional analysis of the HPT axis or measuring the activity of key enzymes. This makes the mechanistic hypothesis overly simplistic, lacking the robust experimental data necessary to substantiate these claims.

We thank the reviewer for their feedback. We recognize that we have not specifically investigated hypothalamic and/or pituitary hormones or their enzyme changes. As such we have amended the manuscript to describe better reflect that our data only indicates potential dysregulation at the level of the testis (Please see lines 843 - 851). While outside the scope of this paper, our future work will look to include a more detailed investigation on the impact of PFAS treatment on the wider HPT axis.

5. Although the study observed that low-dose PFAS exposure had a more pronounced effect on hormone levels than high-dose exposure, the explanation for this phenomenon remains superficial, relying on the literature's reference to "nonlinear dose-response" patterns. There is no in-depth discussion of why this dose-response relationship is so marked in this study, leaving the interpretation of these results incomplete and somewhat superficial.

Answer: Thank you for this important comment. We agree that more pronounced effect on hormone level achieved following the administration of a low- versus high-dose PFAS cocktail was an unexpected result and is not one for which we have any direct data to identify the biological mechanism. However, based on a thorough literature review of toxicological data, such findings are not without precedent. For instance, it has been reported that cognitive function in older adults (PMID: 33068581) and the prevalence of diabetes (PMID: 36964247 and PMID: 35970987) both exhibit a non-monotonic dose-response with plasma PFAS. In the absence of experimental data to substantiate mechanisms of action, we had refrained from speculating on the phenomenon of why this dose-response relationship is so marked. However, in view of this comment and that raised by Reviewer 1 (see Reviewer 1, Question 2 above), we have now taken the opportunity to amend the Discussion to include additional context and a brief consideration of possible explanations. Please see lines 782-789.

6. The study hypothesizes that PFAS exposure may affect offspring health and development via alterations in sperm sncRNA, but no experiments were conducted to directly test these transgenerational effects. This weakens the argument, as the conclusion about potential transgenerational impacts is speculative and needs further experimental evidence to be convincingly established.

Answer: As stated above in response to Reviewer 2 (see Question 3 above), we readily acknowledge that embryonic / offspring phenotypic data would be beneficial in terms of

validating our conclusions. However, we agree with the Editor that that such detailed analyses regrettably fall beyond the scope of the present study. In recognition of the importance of these analyses, we are currently designing a separate study to investigate the impact of PFAS exposure across multiple generations, which would enable us to conduct the suggested in-depth phenotypic analyses of offspring health across each generation. In acknowledgment of these limitations, we have revised our Conclusion to indicate that further studies and additional phenotypic assessments are necessary to validate the functional consequences of the observed molecular changes in sperm and embryos. Please see lines 925- 929.

7. Inconsistent Figure Sizing and Table Formatting: Figure sizes, such as in Figure 8, lack consistency, which affects the overall presentation quality of the manuscript. Additionally, several tables contain incomplete information, which hinders clarity. The authors should ensure consistent figure sizing and optimize the layout of the tables to present a more polished and comprehensive manuscript.

Answer: Thank you for identifying these issues relating to Figure and Table formatting. These issues appear to have arisen during the generation of the combined manuscript files by the submission portal. We wish to assure the reviewer that all figures and tables were prepared with consistent formatting, and we will ensure this is addressed in the final version of the manuscript.

Jun Wei Pek, PhD
Editorial Board Member
Communications Biology

MS#: COMMSBIO-24-5715-T

RE: Response to Reviewers

Dear Professor Pek,

On behalf of my co-authors, I wish to thank you and the Reviewers for recognizing the merits of our study and providing the opportunity to address the Reviewer's comments in a revised version of our manuscript. Below, we provide point-by-point answers to each of the additional reviewer's comments and include the line numbers to highlight the changes that have been made in our manuscript to address their concerns. As requested in the final revision instructions, we have not submitted a marked-up copy with tracked changes. However, should the editor wish to see this, it is prepared and can be readily provided. We remain committed and enthusiastic to publish this study in Communications Biology, and we hope that you will view our revised article with equal enthusiasm.

All best wishes,

Jacinta

RESPONSE TO REVIEWERS

EDITORIAL REQUESTS

Please see the attached document for editorial requests for the final version (.docx file). Please ensure a completed version of this file is uploaded as a Related Manuscript with your final submission. Please review our final submission file checklist to ensure all necessary files are present with your final submission and to avoid delays in accepting your manuscript. For your reference, a style and formatting guide is available here and includes all of our style requirements.

Answer: We remain committed and enthusiastic about publishing this study in Communications Biology. Accordingly, we have also adhered to the attached document and revision checklist guide in relation to manuscript formatting and compliance with journal policies.

REVIEWERS' COMMENTS:

Reviewer #1 (Remarks to the Author):

1. The authors defend GAPDH by normalizing it to tubulin/HSPA2 and conclude all are stable. But if PFAS (or capacitation state) alters multiple housekeeping proteins, cross-normalization can mask concerted shifts; what they need is a total-protein load control or an external spike-in.

Answer: We thank the reviewer for this suggestion. We have now addressed this in two complementary ways. Firstly, we have performed silver staining of SDS-PAGE gels to confirm equivalent total protein loading across all samples (see figure below, panel A; noting that these samples correspond to those reported in our manuscript). Secondly, we used stain-free gel technology from Bio-Rad (see figure below, panel C) to visualize total protein loading on the SDS-PAGE gel prior to subjecting the gel to Western transfer onto a nitrocellulose membrane. The same membrane was then probed with antibodies against the loading control originally reported in our manuscript (i.e., GAPDH), allowing us to confirm that none of the PFAS exposure regimens (i.e., the low or high PFAS dose cocktails used in this study) nor the capacitation status of spermatozoa noticeably affected GAPDH abundance in any of the sperm samples analyzed. In addition, complementary densitometric analysis of the silver-stained gels and those incorporating Bio-Rad stain-free reagents (see figure below, panel B and D, respectively) both support these findings and thereby confirm that normalization using GAPDH is appropriate in our dataset.

Figure 1. Verification of total protein loading and assessment of GAPDH stability in mouse spermatozoa following PFAS exposure of male mice and also in response to the capacitation status of mouse spermatozoa. **(A)** Direct visualization of silver-stained SDS-PAGE gels and **(B)** densitometric analysis of these silver-stained gels confirmed equivalent total protein loading across all samples. **(C)** Bio-Rad stain-free imaging and complementary **(D)** densitometric analysis of total protein loading also confirmed equal protein loading, and that GAPDH abundance in sperm samples was not affected by either PFAS exposure nor the capacitation status of these cells.

2. Please also use nomenclature in the manuscript text and figures—tRNA-derived small RNAs (tsRNAs), rRNA-derived small RNAs (rsRNAs), and mRNA-derived small

RNAs (msRNAs)—in parallel with tRFs and rRFs. Limiting these terms to the response letter for reviewer only reduces discoverability.

Answer: We thank the reviewer for this suggestion and acknowledge the importance of using correct nomenclature to avoid confusion. Accordingly, we have amended our manuscript to include tsRNAs as an alternative abbreviation for tRNA derived small RNAs. This has been included at their first mention in both the text and figure legends. In addition, we have elected to remove the abbreviation of rRNA fragments, in favor of using its full term, as this is only mentioned twice in the text.

We have, however, refrained from referring to mRNA derived small RNAs as 'msRNAs'. Our reasoning is that the msRNA abbreviation has been used in recent literature to refer to both microRNA-size small RNAs in *E.coli* (PMID: 23783561) and medium sized RNAs (PMID: 33404286) but appears to be used sparingly to refer to messenger RNA small RNAs. To avoid risking further confusion, we have chosen to refer to this class of small RNAs as mRNA fragments without introducing an abbreviation."

Please see lines 320, 322–323, 570, and 1138 - 1139. We have also updated the acceptable acronyms for tRNA fragments in the figure legend of Supplementary Table 2. This now reads "tRFs/tDRs/tsRNA = tRNA fragments," with the change highlighted in yellow. The updated version can be found in the supplementary information file.

Reviewer #2 (Remarks to the Author):

1. The authors have adequately addressed previous comments. The revised manuscript is now suitable for publication.

Answer: Thank you very much for taking the time to again review our manuscript and for your kind words. We truly appreciate your thoughtful assessment and are pleased to hear that you consider the revised manuscript suitable for publication.

Reviewer #3 (Remarks to the Author):

1. This paper provides a comprehensive assessment of the impacts of environmentally relevant PFAS mixtures on male reproductive function in mice. The authors present valuable findings on reduced sperm production and altered sperm epigenetics linked to early embryonic gene dysregulation. While some results align with prior studies, the integration of sperm function, epigenetic changes, and embryo development offers novel insights. The conclusions are adequately supported by experimental data, though additional research could further strengthen causal links. The statistical methods are appropriate, and methodological details enable reproducibility, despite minor ambiguities. Overall, this well - written paper contributes significant data to the field of reproductive toxicology and environmental health, highlighting male reproductive vulnerability to PFAS. It has the potential to influence future research and policies, and thus, I recommend its acceptance.

Answer: Thank you very much for your thorough review and for your generous feedback on our manuscript. We greatly appreciate your recognition of the novelty and significance of our findings, as well as your thoughtful evaluation of our methodology and analyses, and we are pleased to hear you consider the work a valuable contribution to the field.